# A chemically induced attenuated strain of *Candida albicans* generates robust protective immune responses and prevents systemic candidiasis development

Swagata Bose, Satya Ranjan Sahu, Abinash Dutta, Narottam Acharya*

Department of Infectious Disease Biology, Institute of Life Sciences, Bhubaneswar, India

*For correspondence:
narottam_acharya@ils.res.in

**Abstract** Despite current antifungal therapy, invasive candidiasis causes >40% mortality in immunocompromised individuals. Therefore, developing an antifungal vaccine is a priority. Here, we could for the first time successfully attenuate the virulence of *Candida albicans* by treating it with a fungistatic dosage of EDTA and demonstrate it to be a potential live whole cell vaccine by using murine models of systemic candidiasis. EDTA inhibited the growth and biofilm formation of *C. albicans*. RNA-seq analyses of EDTA-treated cells (CAET) revealed that genes mostly involved in metal homeostasis and ribosome biogenesis were up- and down-regulated, respectively. Consequently, a bulky cell wall with elevated levels of mannan and β-glucan, and reduced levels of total monosomes and polysomes were observed. CAET was eliminated faster than the untreated strain (Ca) as found by differential fungal burden in the vital organs of the mice. Higher monocytes, granulocytes, and platelet counts were detected in *Ca-* vs CAET-challenged mice. While hyper-inflammation and immunosuppression caused the killing of *Ca*-challenged mice, a critical balance of pro- and anti-inflammatory cytokines-mediated immune responses are the likely reasons for the protective immunity in CAET-infected mice.

## eLife assessment

This study presents a **useful** strategy in which the authors devised a simple method to attenuate *Candida albicans* and deliver a live whole-cell vaccine in a mouse model of systemic candidiasis. The reviewers are not convinced about the completeness of the study: the strength of the evidence is **incomplete** and could be augmented with additional experiments to more fully characterize vaccine efficacy and host immune responses.

## Introduction

*Candida* species colonizes several anatomical niches of a host, either as a commensal or an opportunistic fungal pathogen. Infections by *Candida* sp. range from superficial oral and vulvovaginal candidiasis (VVC) to serious invasive candidiasis (*Brown et al., 2012*; *Peroumal et al., 2022*). In immunosuppressed conditions, these fungi can evade subsided mucosal immunity and disseminate into the bloodstream, resulting in systemic candidiasis with a mortality rate of >40%. *Candida albicans* is the frequently isolated species responsible for the majority of these infections, thus, imposing a huge threat to human health worldwide. Currently, polyenes, echinocandins, and azoles are the three major

classes of antifungals available to combat fungal diseases (*Ford et al., 2015*; *Lee et al., 2021*). These drugs are characterized by a narrow spectrum of activity to target limited fungal species and often exhibit cytotoxicity to cause side effects. Over 1.5 million yearly global burdens of lethal infections suggest that the treatment of fungal diseases by using available antifungal drugs seems to be partially helping, and additionally, the emergence of drug-resistant isolates clearly emphasize the urgent need of better diagnostics, safe and broad-spectrum novel antifungal drugs, and effective immunotherapeutics and prophylactics (*Rudkin et al., 2018*). Since no approved vaccine is available for human use, designing and developing vaccines against fungal pathogens is a priority (*Sahu et al., 2022*). NDV-3A and PEV7 are two recombinant vaccines that seem to be protecting recurrent VVC during initial human clinical trials (*De Bernardis et al., 2012*; *Edwards et al., 2018*). In addition, various studies on host-fungal interactions have helped to identify multiple experimental vaccine candidates and those were shown to be safe and effective in animal models (*Sahu et al., 2022*). Most of these are suggested to generate neutralizing antibodies and are expected to be effective against *Candida* infections. The interaction between the host immunity and fungal pathogens is very complex and varies depending upon the nature and site of infections, and there are at least three layers of immune responses operating simultaneously against the invaders: innate, adaptive, and trained (*Netea et al., 2015*; *Romani, 2004*). Also, as *C. albicans* is a multimorphic fungus, designing and developing a whole-cell vaccine approach seems to be arguably ideal.

Whole-cell vaccines can be either live or killed, and both kinds of vaccines are effectively in use against viral and bacterial infections. Heat-killed *C. albicans* and *Saccharomyces cerevisiae* strains were found to be partially protecting against experimental vaginal candidiasis (*Cárdenas-Freytag et al., 2002*; *Liu et al., 2012*). Some of the genetically engineered *C. albicans* strains (tet-NRG1, PCA-2, CM1613, etc.) were reported to be attenuated and protective against disseminated candidiasis in mice (*Sahu et al., 2022*; *Saville et al., 2009*). However, clinical trials for these strains are yet to be commenced. Since *C. albicans* co-evolves with the host and lives mostly as a beneficial gut pathogen (*Peroumal et al., 2022*), the host must be employing certain mechanisms to block its transition to a pathogenic trait which can be explored to develop attenuated strains of *C. albicans* naturally rather than using a recombinant DNA technology approach. 'Nutritional immunity' is one such frontline defense mechanism that limits fungal growth and pathogenesis (*Hood and Skaar, 2012*).

Metals play a pivotal role in host-pathogen interaction. Metals like iron, zinc, manganese, magnesium, copper, etc. being a cofactor of various metabolic and non-metabolic enzymes are essential for both host and pathogens. Therefore, excessive or limited availability of metals is deleterious to cells. Several studies suggest that the virulence and pathogenesis of fungi require a threshold level of these metals (*Crawford and Wilson, 2015*; *Gerwien et al., 2018*). To starve pathogens of essential metals, the host organism employs nutritional immunity strategies to reduce the availability of metals, thereby the virulence and disease progression of a pathogen is suppressed drastically (*Almeida et al., 2008*; *Bates et al., 2005*; *Citiulo et al., 2012*). Our recent report found that a unknown soluble factor released from *E. coli* during co-culturing inhibited the growth, biofilm formation, and morphological transition from yeast to hyphae of *C. albicans* (*Bose et al., 2023*). In an animal model, the culture supernatant of *E. coli* attenuated the virulence of *C. albicans* significantly. Chemical complementation analyses revealed that the exogenous addition of divalent metals rescued the growth of *C. albicans* which was inhibited by *E. coli* (*Bose et al., 2023*). Based on the concept of nutritional immunity and these reports, we argued that the metal chelators like EDTA, DTPA, etc. could mimic *E. coli* in vitro to regulate the virulence and pathogenesis of *C. albicans*. Live whole-cell vaccine strains are generated mainly by genetic manipulation, here, we developed and characterized for the first time a chemical reagent-based attenuated strain of *C. albicans* which is safe and effectively protects systemic candidiasis in a pre-clinical model.

## Results
### Metal chelators inhibit the growth and biofilm of *C. albicans*

To begin with, the effect of a range of concentrations of EDTA (0.9–500 µM) on the growth of *C. albicans* was determined by measuring the optical density ($OD_{600nm}$) of the culture in the presence of the reagent. Although we observed a significant growth retardation of *C. albicans* at 125 µM EDTA, upon addition of 250 µM EDTA, the growth was inhibited completely and remained saturated beyond

this concentration. This result was also validated on solid YPD media where no colony formed at the highest dilution of the spot when the *C. albicans* was treated with 250 µM EDTA (*Figure 1—figure supplement 1A—C*). Henceforth, 250 µM EDTA was used for subsequent analyses. To determine the effect of EDTA on the survivability of fungal cells, *C. albicans* cells ($OD_{600nm}$ = 0.5 or $10^7$ cells per mL) in YPD media were grown for 12 hr without and with 250 µM of the metal chelator. At various time intervals, an equal number of cells were harvested and stained with SYTOX Green and propidium iodide (PI), and analyzed by flow cytometry and microscopy, respectively (*Figure 1—figure supplement 2A—D*). Both analyses suggested that 250 µM EDTA had minimal effect on the survivability of *C. albicans* cells during this period of growth as an equal percentage of dead cells were detected in untreated as well as treated conditions. Thus, the inhibitory effect of 250 µM EDTA on the growth of *C. albicans* cells is not due to the accumulation of dead cells but rather due to the slowing down of the cell division process. To further characterize the effect of EDTA, we checked various morphological forms of *C. albicans* upon exposure to 250 µM EDTA and 10% serum (*Figure 1—figure supplement 3A—C*). *C. albicans* exists predominantly in the round form under normal physiological conditions; however, in the presence of inducers like serum, it transits into its reversible forms such as pseudo-hyphal and hyphal forms. Our results revealed that the 250 µM EDTA treatment did not alter the morphology of *C. albicans* cells and it minimally affected the morphological switch induced by serum. Even the germ tube length induced by serum was barely affected by the addition of EDTA at this concentration.

Since EDTA inhibited the *C. albicans* growth, to identify the metals that could effectually rescue *C. albicans* survival, we supplemented various divalent metal salts to the EDTA-treated *C. albicans* cells (*Figure 1A and B*). Both the growth curve and CFU analyses revealed that the antagonizing effect of 250 µM EDTA on *C. albicans* was abolished by supplementing with metals like $MgSO_4$, $MnCl_2$, $FeCl_2$, and $ZnCl_2$. Since the number of colonies remained the same for 8–24 hr of treatment with EDTA, it again suggested that EDTA most likely is a fungistatic inhibitor whose effect can be reversed by divalent metal complementation. To further strengthen the result, $MgSO_4$ was supplemented to the EDTA treated culture at 6 hr post-treatment rather than at 0 hr and growth was allowed. Interestingly, the suppressed growth of EDTA-treated *C. albicans* cells was rescued significantly and the effect 250 µM EDTA was minimized even in this context (dark blue line). Interestingly, the exogenous addition of these metals (8 µM) alone to the YPD media had no or minimal effect on the growth of *C. albicans*.

To strengthen the result, the effect of EDTA and metals on the preformed biofilm of *C. albicans* was determined (*Figure 1C and D*). Consistent with the growth curve result, EDTA inhibited the biofilm of *C. albicans* but was rescued by the addition of divalent metals. To substantiate the inhibitory role of EDTA, the effect of four more metal chelators Diethylene triamine pentaacetic acid (DTPA), *N,N=,N=-*tetrakis(2-pyridylmethyl)ethane-1,2-diamine (TPEN), Aprotinin, and Ciclopirox ethanolamine (CE) on *C. albicans* growth was determined. DTPA is an aminopolycarboxylic acid consisting of a diethylene-triamine backbone with five carboxymethyl groups, and has a high affinity for di- and tri-metal cations (*Polvi et al., 2016*). TPEN is an intracellular membrane-permeable ion chelator of transition metals (*Cho et al., 2007*). CE acts by binding to and chelating trivalent cations mainly iron (*Niewerth et al., 2003*). Unlike these chelators, BPTI or aprotinin is an antimicrobial peptide that lowers the cellular levels of magnesium (*Bleackley et al., 2014*). Growth curve, CFU analysis, and biofilm formation assays revealed that all of these metal chelators inhibited the growth phenotypes of *C. albicans* with varied efficiency (*Figure 1E–H*). While the effects of DTPA and aprotinin were very similar to EDTA; TPEN and CE inhibited both growth and biofilm of *C. albicans* drastically even at a lower concentration (100 µM) and accordingly, hardly any colony formed on solid YPD plates, and poor biofilm was observed. These results suggested that essential metals like magnesium, manganese, iron, and zinc are required for the growth and biofilm development of *C. albicans*, and sequestration of these by a range of metal chelators affects the growth of *C. albicans*.

## Global transcriptomic changes of *C. albicans* in response to EDTA treatment

Next, we investigated any changes in *C. albicans* gene expression due to EDTA treatment using high-throughput RNA sequencing. To achieve our goal, total RNA isolated from *C. albicans* cells grown for 6 hrs without or with 250 µM EDTA treatment was subjected to RNA sequencing using an Illumina NovaSeq platform, and a summary of reads obtained from each sample and their mapping to the

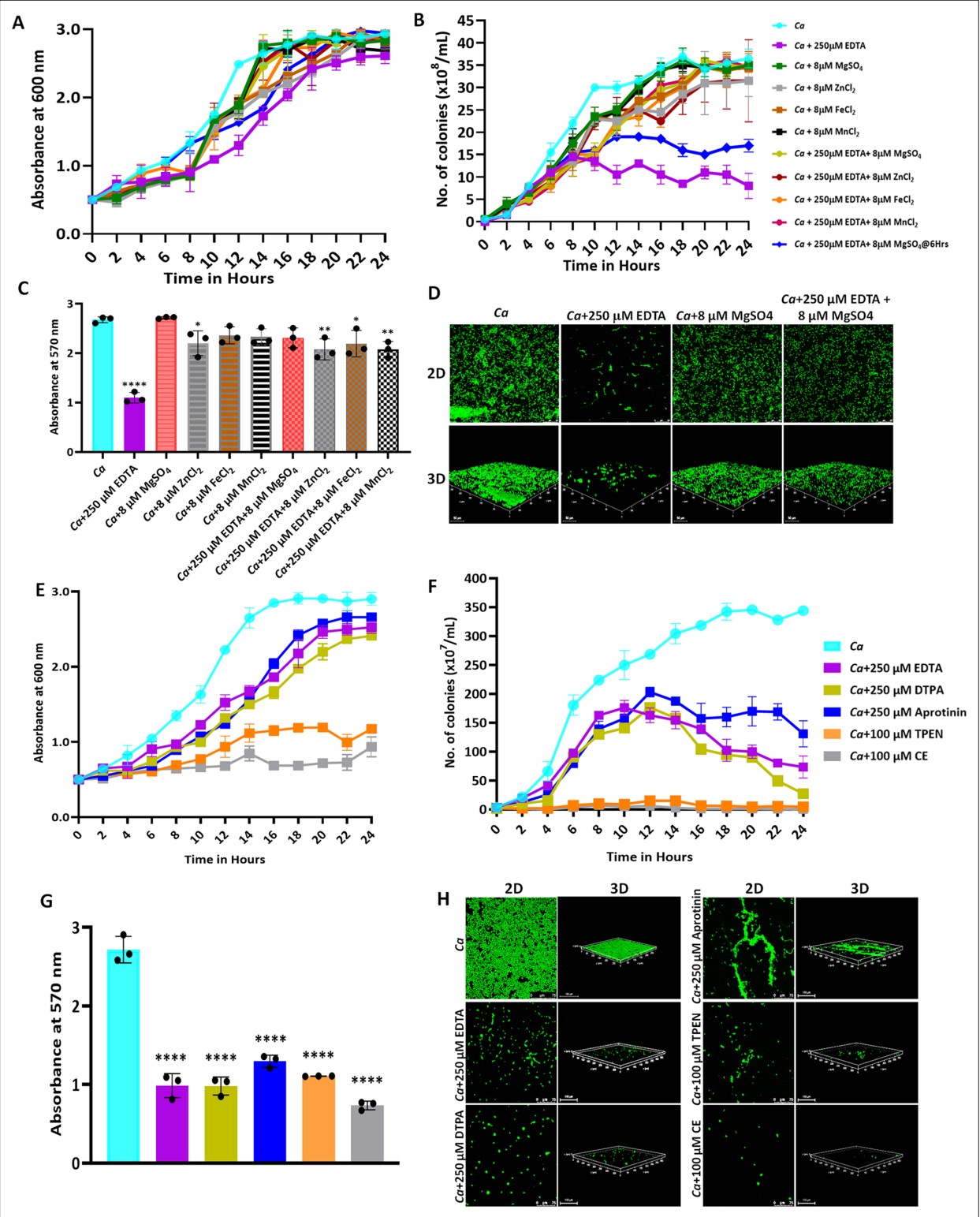

**Figure 1.** Effect of metal chelators on the growth and biofilm of *C. albicans*. (**A**) *C. albicans* cells were cultured in YPD media at 30 °C for 24 hr without (cyan blue) and with the indicated concentration of EDTA (purple), MgSO₄ (Green), ZnCl₂ (Grey), FeCl₂ (Brown), and MnCl₂ (Black), EDTA+MgSO₄ (Lime green), EDTA+MgSO₄@6 hr (dark blue), EDTA+ZnCl₂ (Maroon), EDTA+FeCl₂ (Orange), and EDTA+MnCl₂ (Pink). Optimal absorbance was measured at 600 nm in different intervals of incubation. (**B**) The cultures from the above-mentioned experiment was diluted and plated on YPD plate. Colonies were counted and plotted to determine the CFU efficiency. (**C**) Pre-formed *C. albicans* biofilm was treated with EDTA and divalent metals as mentioned. Their effect on biofilm was observed post-24 hr treatment by crystal violet staining and estimating at 570 nm. (**D**) Effect of EDTA and divalent metals on *C.*

*Figure 1 continued on next page*

*Figure 1 continued*

*albicans* biofilm was again observed by acridine orange staining and visualization under a ×40 magnification using a CLSM. Similarly, the effect of other metal chelators like DTPA (Lime green), Aprotinin (blue), TPEN (orange) and CE (grey) on the growth (**E**), CFU (**F**), and biofilm formation of *C. albicans* analyzed by crystal violet staining (**G**) and CLSM (**H**). Mean values from three independent experiments considered and error bar represents SEM. p values *<0.05, **<0.01 and ****<0.0001 were significant as determine by one-way ANOVA.

The online version of this article includes the following source data and figure supplement(s) for figure 1:

**Source data 1.** Effect of metal chelators on the growth, CFU, and biofilm of *C. albicans*.

**Figure supplement 1.** Identification of a fungistatic concentration of EDTA.

**Figure supplement 2.** Effect of 250 µM EDTA on cell viability.

**Figure supplement 3.** Effect of EDTA on *C.albicans* morphology.

reference genome is given in *Table 1*. On average, we obtained 19,190,581 reads per sample with 98% or more successfully mapping to the known mRNA in the *C. albicans* strain SC5314 reference genome (assembly ASM18296v3). Principal-component analysis (PCA) was conducted to provide a pictorial representation of the transcriptomic similarities among various triplicates and it revealed that the EDTA-treated *C. albicans* samples clustered separately from the untreated ones, indicating a high level of correlation among samples with distinct transcriptome profiles (*Figure 2—figure supplement 1A*). The total differentially expressed genes in untreated and EDTA-treated *C. albicans* cells (here onwards it will be named as *Ca* and CAET, respectively) were 5915, of these, 3093 and 2822 genes were up and downregulated, respectively. So, our RNA sequencing data indicated that EDTA has a profound effect on *C. albicans* gene expression resulting in alterations in the transcriptome. In order to find genes that were significantly dysregulated, two cut-offs: (1) false discovery rate (FDR) cut-off ≤0.05 and (2) log2 fold change (FC) cut-off ≥1 were applied; and a total of 799 differentially expressed genes (DEGs), of which 411 were upregulated and 388 were downregulated in CAET. The volcano plot depicted two pools of genes that are significantly up- (red dots) and down – (blue dots) regulated in CAET (*Figure 2A*). The hierarchical clustering algorithm was used to create a heat map by taking top 25 up- and 25 down-regulated DEGs (*Figure 2B*). The heat map suggested a minimal variation in cluster linkage among similar samples and thus, the dysregulated DEGs upon treatment with EDTA relative to the untreated controls were consistently observed. We re-verified the RNA seq results by carrying out real-time and semi-quantitative PCR analyses of representative genes. Upregulation of *ZRT1*, *CHT4*, and *PGA13*, and downregulation of *ACO2*, *PLB1*, and *THI13* genes were observed in CAET in comparison to *Ca* (*Figure 2—figure supplement 1B*).

## Treatment with EDTA upregulates the expression of *C. albicans* genes required mainly in metal transport and pathogenesis

In the analysis of the top 100 upregulated genes those expressions varied >fivefolds higher in response to EDTA (*Table 2*, *Figure 2C*), about 57% of uncharacterized genes with unknown functions emerged. Interestingly, the function of 6 out of 12 top most upregulated genes that were >30 folds more expressed in treated than the untreated *C. albicans* cells are yet to be deciphered. UniProt only predicts the following uncharacterized proteins CAALFM_C200860CA (orf19.9586 may encode a leucine-rich repeat protein of 418 aa with protein kinase binding function), CAALFM_C401340WA (orf19.12123

**Table 1.** A brief summary of RNA sequencing reads obtained from Illumina Novaseq.

| Sample Name (Sequencing ID) | Read Length (bp) | Total raw reads | Reads after rRNA removal | Average Length after trim | Read after trimming | % known mRNA | % Unknown mRNA |
|---|---|---|---|---|---|---|---|
| Wild type (MT523) | 151×2 | 21,937,416 | 19,222,356 (87.62%) | 143.22 | 19,221,728 | 98.06 | 1.94 |
| Wild type (MT524) | 151×2 | 20,794,992 | 18,475,070 (88.84%) | 141.71 | 18,474,428 | 98.36 | 1.68 |
| Wild type (MT525) | 151×2 | 20,564,218 | 17,699,794 (86.07%) | 140.89 | 17,699,180 | 98.3 | 1.7 |
| CAET (MT526) | 151×2 | 22,376,466 | 20,284,442 (90.65%) | 140.79 | 20,283,776 | 98.15 | 1.85 |
| CAET (MT527) | 151×2 | 21,412,804 | 20.073,926 (97.49%) | 142.14 | 20,073,230 | 98.72 | 1.28 |
| CAET (MT528) | 151×2 | 20,435,938 | 19,391,726 (94.89%) | 142.25 | 19,391,142 | 98.85 | 1.5 |

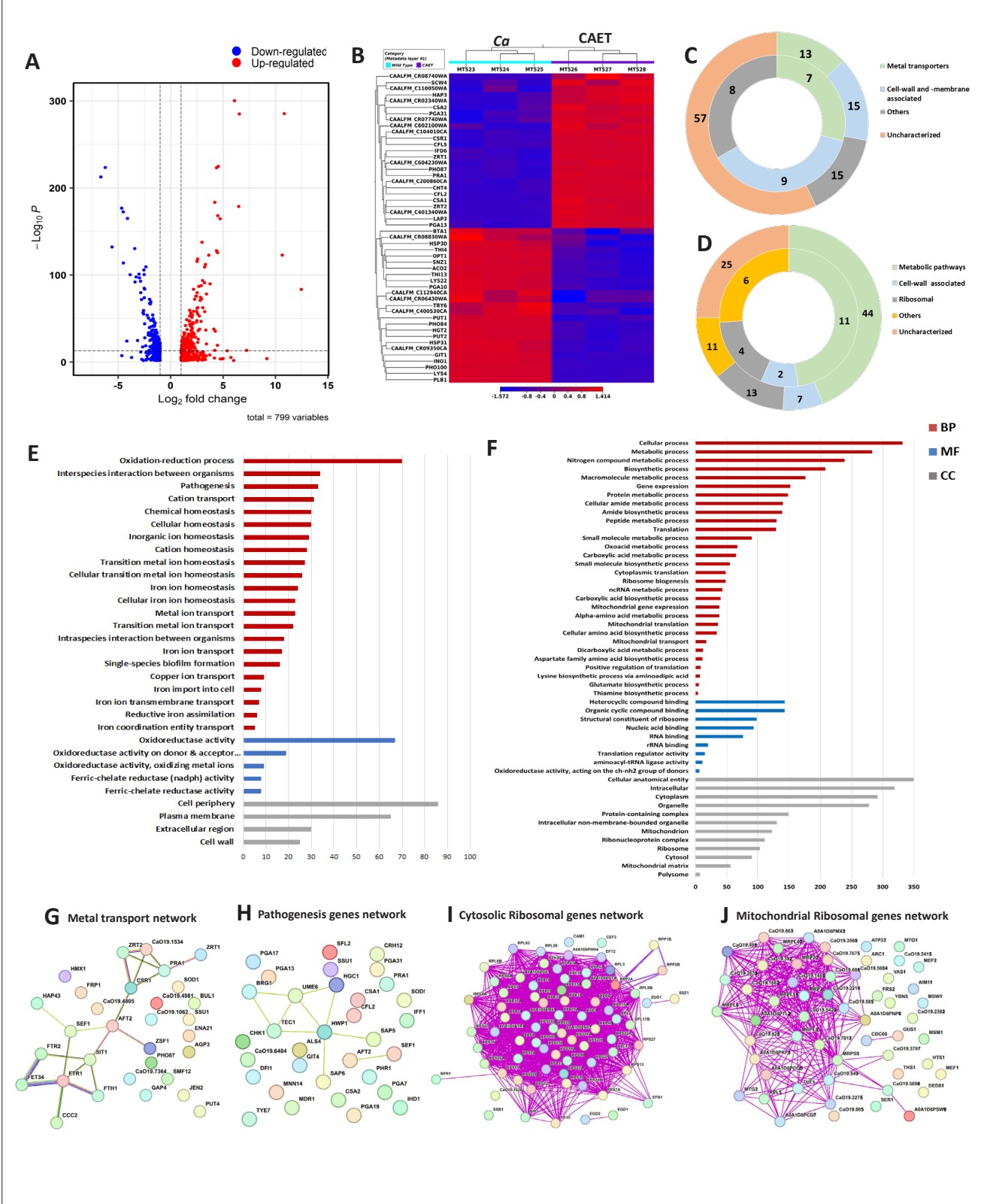

**Figure 2.** Transcriptomics analyses *C.albicans* cells upon EDTA treatment. (**A**) A Volcano plot depicting differentially expressed genes (DEGs) in CAET cells. The red dots indicate upregulated genes (411) and blue dots indicates the downregulated genes (388). (**B**) Heat map showing relative abundance of top 25 significantly upregulated and downregulated genes in *Ca* and CAET. RNA sequencing was carried out in triplicates. Red colour indicates upregulation and blue indicates downregulation as denoted by the Z score. (**C**) Double donut chart indicates top 100 significantly upregulated genes

*Figure 2 continued on next page*

*Figure 2 continued*

categorized into four main groups that is, metal transporters (light green), cell wall- and membrane-associated genes other than metal transporters (blue), others (includes drugs resistance-associated, morphology-associated, biosynthetic and catalytic processes; represented in grey), and uncharacterized genes (cream). Inner donut depicts the number of DEGs from the above categories associated with virulence and pathogenesis in *C. albicans* as reported by published literatures. (**D**) Double donut chart depicting top 100 significant downregulated gene categorized as metabolic pathways (green), cell-wall associated (blue), ribosomal (grey), others (includes morphology-associated, resistance-associated; represented in yellow), and uncharacterized (cream). Outer circle represents the total number of genes in each category and inner circle indicates the virulence-related genes out of the total number of genes in that specific category. (**E**) Gene ontology enrichment analysis for 411 upregulated DEGs. (**F**) Gene ontology enrichment analysis for 388 downregulated DEGs were plotted. Both in (**E**) and (**F**), *x*-axis represents the number of DEGs and *y*-axis represents various processes like BP (biological processes), MF (molecular functions), and CC (cellular components) indicated in maroon, blue, and grey colors, respectively. (**G**) STRING analyses showing relationship between DEGs. Interaction amongst 15 connected out of 31 upregulated DEGs involved in metal transport and homeostasis is shown. (**H**) STRING analysis shows 9 connected out of 33 upregulated DEGs involved in pathogenesis. (**I**) Among the down-regulated DEGs, 70 connected out of 74 DEGs belonging to cytosolic ribosomal genes as determined by STRING. (**J**) 38 connected out of 56 DEGs belonging to mitochondrial ribosomal components were connected by as determined STRING.

The online version of this article includes the following source data and figure supplement(s) for figure 2:

**Source data 1.** Transcriptomic analyses of CAET.

**Figure supplement 1.** Transcriptomic analyses of CAET and validation.

may encode a secreted protein of 109 aa), CAALFM_CR08740WA (orf19.13808 may encode a SWIM-type domain-containing protein of 226 aa), CAALFM_CR02340WA (orf19.11220 may encode a vacuolar protein sorting-associated protein 8 homolog of 419 aa), CAALFM_CR07740WA (orf19.2633.1 may encode a Histone H2B protein of 64 aa), and CAALFM_C602100WA (orf19.10993/*LDG8* may encode a cell wall protein of 167 aa) (***Supplementary file 1***). We found 13 DEGs related to metal transport and homeostasis, 15 DEGs of cell wall and membrane-associated proteins other than functions related to metals, and another 15 DEGs of diverged functions. The characterized top 6 upregulated genes with >30 folds higher expression were *ZRT1* (zinc ion transmembrane transported required for zinc sequestration), *PRA1* (cell surface protein involved in zinc binding and metallopeptidase activity), *CFL5* (Ferric-chelate reductase required for iron and copper ions uptake), *CFL2* (Ferric-chelate reductase required for iron and copper ions uptake), *PHO87* (SPX domain-containing inorganic phosphate transporter), and *ZRT2* (zinc ion transmembrane transporter activity) having roles in zinc, iron, and phosphate transportation (***Citiulo et al., 2012***; ***Lan et al., 2004***; ***Wilson et al., 2012***). In the category of <30 but >~fivefolds higher expressed genes, 51 uncharacterized genes, 7 genes (*CSA1, CSA2, CSR1, HAP3, CFL4, FRE9*, and *ARH2*) having a role in essential metal transport and homeostasis, 15 cell wall and plasma-membrane-associated genes (*SCW4, PGA13, PGA31, CHT4, RIM9, ALS4, CSH1, MRV8, HGT5, CDA2, PRN4, GLX3, PGA7, RBT5*, and *JEN1*), and 15 genes (*LAP3, IFD6, ATO1, NOP6, RAS2, HST6, STE23, FGR46, POL93, HPD1, AOX2, SOU2, GPD1, CAG1*, and *FDH1*) with other function. More importantly, according to earlier studies, 24 of these upregulated genes were shown to be associated with fungal virulence, thus, EDTA-treated *C. albicans* cells might exhibit differential pathogenicity (***Gerwien et al., 2018***; ***Kuznets et al., 2014***; ***Wilson et al., 2012***; ***Zheng et al., 2004***).

Further, GO annotation analyses of all 411 upregulated genes revealed a total of 37 GO terms 28 biological processes (BP), 5 molecular functions (MF), and 4 cellular components (CC) that were enriched upon EDTA treatment (***Supplementary file 2A***, ***Figure 2E***). The highly significant upregulated biological processes GO terms were oxidation-reduction (GO:0055114), inter-species interaction (GO:0044419), pathogenesis (GO:0009405), and transition metal transporters and utilization (GO:0006812, GO:0048878, GO:0019725, GO:0098771, etc.). Similarly, GO terms of molecular function (GO:0098771, GO:0016616, etc.) genes encode for enzymes required for oxidation-reduction reactions, and GO terms of cellular component (GO:0071944, GO:0005886, etc.) mostly indicate upregulated genes belong to the cell wall and its periphery. To find out the gene networks that were affected upon EDTA treatment, STRING (Search Tool for the Retrieval of Interacting Genes/Proteins) analysis was carried out (***Supplementary file 3A***). High confidence (0.7 confidence level) analysis revealed upregulation of network proteins mostly involved in metal ion transport and 18 strong nodes were detected among 15 out of 31 genes (***Figure 2G***). Another gene network of 33 genes that caught our attention involved in pathogenesis and 9 DEGs of this network are directly or indirectly shown to be involved in fungal pathogenesis (***Supplementary file 4***, ***Figure 2H***). Altogether, our analysis suggested that the upregulation of cell wall and plasma membrane-associated genes whose

**Table 2.** List of top 100 upregulated genes in EDTA treated *C. albicans* cell (CAET).

**Metal Transporters**

| No. of DEGs | Name of DEGs (13) | Fold Change | FDR p-value | Function |
|---|---|---|---|---|
| 1 | ZRT1 | 5644.9942 | 5.20999E-82 | Zinc ion transporter; Hyphal inducer |
| 2 | PRA1 | 1829.3467 | 1.3833E-282 | Zinc sequester; pH regulated antigen |
| 3 | CFL5 | 151.1291 | 4.16014E-13 | Ferric reductase |
| 4 | CFL2 | 94.2481 | 1.5889E-282 | Ferric reductase; Virulence |
| 5 | PHO87 | 90.3067 | 1.0754E-176 | Phosphate transporter; Virulence |
| 6 | ZRT2 | 68.1720 | 2.5856E-297 | Zinc uptake; Biofilm inducer |
| 7 | CSA2 | 26.6647 | 0.000886698 | Hyphal inducer; Heme utilization protein; Biofilm inducer |
| 8 | CSR1 | 23.2254 | 2.0725E-222 | zinc homeostasis, Filamentation inducer; |
| 9 | CSA1 | 22.5038 | 3.4486E-166 | Hyphal inducer; Iron homeostasis; Biofilm inducer; |
| 10 | HAP3 | 18.4511 | 5.412E-13 | Iron homeostasis; |
| 11 | CFL4 | 8.4663 | 3.22997E-11 | Uptake of iron metal ion |
| 12 | FRE9 | 7.5052 | 1.41623E-45 | Ion transport, Ferric reductase |
| 13 | ARH2 | 5.7589 | 1.18159E-60 | Involved in heme biosynthesis |

Cell-wall and -membrane associated

| | Name of the DEGs (15) | Fold Change | FDR p-value | Function |
|---|---|---|---|---|
| 1 | SCW4 | 28.9947 | 0.000794714 | Glucanase activity |
| 2 | PGA13 | 26.3347 | 8.3797E-163 | Cell wall integrity; Morphogenesis, Virulence |
| 3 | PGA31 | 22.0969 | 1.25365E-18 | Cell wall integrity |
| 4 | CHT4 | 20.7121 | 4.5927E-126 | Chitin synthesis and degradation |
| 5 | RIM9 | 12.4158 | 4.4708E-121 | Promote growth under alkaline condition |
| 6 | ALS4 | 10.0218 | 1.4683E-110 | Enhance cell surface adhesion |
| 7 | CSH1 | 9.5387 | 1.2788E-107 | Maintain cell integrity; Virulence |
| 8 | MRV8 | 8.7526 | 0.007934383 | Biofilm inducer; Caspofungin tolerance |
| 9 | HGT5 | 7.5433 | 2.42539E-86 | Glucose transporter |
| 10 | CDA2 | 7.2664 | 1.39323E-07 | Chitin deacetylase |
| 11 | PRN4 | 6.2037 | 5.00644E-35 | Protein with similarity to pirins |
| 12 | GLX3 | 5.9890 | 3.73497E-61 | Binds human IgE; Induce biofilm |
| 13 | PGA7 | 5.6247 | 2.14119E-46 | GPI-linked hyphal surface antigen; Induce spider biofilm |
| 14 | RBT5 | 5.5414 | 7.36268E-45 | GPI-linked cell wall protein; utilize haemin and haemoglobin; biofilm induced |
| 15 | JEN1 | 5.1930 | 4.63735E-06 | Localized on plasma membrane and induced by lactic acid |

Others

| | Name of the DEGs (15) | Fold Change | FDR p-value | Function |
|---|---|---|---|---|
| 1 | LAP3 | 20.7648 | 6.7063E-221 | Aminopeptidase; Bleomycin resistance |
| 2 | IFD6 | 18.4736 | 2.2359E-181 | Azole resistance; Biofilm-inducer |
| 3 | ATO1 | 13.3199 | 6.23932E-08 | pH neutralizer in macrophage phagolysosome |
| 4 | NOP6 | 10.5223 | 3.90346E-07 | Ribosome biosynthesis |
| 5 | RAS2 | 10.2667 | 8.69395E-36 | Filamentous growth enhancer; cAMP pathway regulator |

*Table 2 continued on next page*

*Table 2 continued*

**Metal Transporters**

| | | | | |
|---|---|---|---|---|
| 6 | HST6 | 8.6960 | 6.13855E-09 | Transport lipid; antifungal drugs resistance; Encode ABC transporters |
| 7 | STE23 | 8.3030 | 3.41151E-92 | Involved in processing of mating pheromones |
| 8 | FGR46 | 7.4482 | 1.72544E-07 | Filamentous Growth Regulator |
| 9 | POL93 | 7.0291 | 1.77132E-69 | Encode reverse transcriptase, protease and integrase; Induce biofilm |
| 10 | HPD1 | 5.9926 | 6.53346E-22 | Involved in degradation of toxic propionyl-CoA; Induce spider biofilm |
| 11 | AOX2 | 5.6936 | 7.2209E-114 | Induce biofilm formation |
| 12 | SOU2 | 5.5746 | 0.006421918 | L-sorbose utilization; biofilm induced |
| 13 | GPD1 | 5.4805 | 2.01281E-74 | Involved in glycerol biosynthesis |
| 14 | CAG1 | 5.3023 | 2.20049E-05 | Role in mating pheromone response |
| 15 | FDH1 | 5.2775 | 2.01281E-74 | xidization of formate to $CO_2$ |

**Uncharacterized**

| | Name (57 DEGs) | Fold Change | FDR p-value | Function |
|---|---|---|---|---|
| 1 | CAALFM_C200860CA | 1600.2301 | 4.4708E-121 | Uncharacterized |
| 2 | CAALFM_C401340WA | 583.6099 | 0.000537249 | Uncharacterized |
| 3 | CAALFM_CR08740WA | 64.8421 | 0.035751787 | Uncharacterized |
| 4 | CAALFM_CR02340WA | 53.5609 | 3.38443E-05 | Uncharacterized |
| 5 | CAALFM_CR07740WA | 42.2094 | 1.62356E-12 | Uncharacterized |
| 6 | CAALFM_C602100WA | 32.1762 | 0.000435707 | Uncharacterized |
| 7 | CAALFM_C104010CA | 21.9880 | 4.8225E-124 | Uncharacterized |
| 8 | CAALFM_C604230WA | 19.1498 | 1.67518E-34 | Uncharacterized |
| 9 | CAALFM_C110050WA | 18.1320 | 0.000124176 | Uncharacterized |
| 10 | CAALFM_CR06510WA | 17.4047 | 6.21185E-37 | Uncharacterized |
| 11 | CAALFM_CR06500CA | 13.6055 | 2.25687E-88 | Uncharacterized |
| 12 | CAALFM_CR08310CA | 10.6593 | 4.21036E-20 | Uncharacterized |
| 13 | CAALFM_CR06570CA | 10.6284 | 6.6067E-77 | Uncharacterized |
| 14 | CAALFM_C204430WA | 9.9253 | 0.003875235 | Uncharacterized |
| 15 | CAALFM_C303570CA | 9.6111 | 5.27224E-07 | Uncharacterized |
| 16 | CAALFM_C306040WA | 9.4663 | 1.09545E-56 | Uncharacterized |
| 17 | CAALFM_C106860WA | 9.4073 | 0.000100814 | Uncharacterized |
| 18 | CAALFM_C602480WA | 9.3877 | 1.58803E-89 | Uncharacterized |
| 19 | CAALFM_C302790WA | 9.1272 | 2.43382E-06 | Uncharacterized |
| 20 | CAALFM_C503430WA | 8.6904 | 0.000101716 | Uncharacterized |
| 21 | CAALFM_C402930WA | 8.6463 | 2.18312E-11 | Uncharacterized |
| 22 | CAALFM_C403340CA | 8.6294 | 0.000183681 | Uncharacterized |
| 23 | CAALFM_C701010WA | 8.4791 | 3.30179E-61 | Uncharacterized |
| 24 | CAALFM_C202580WA | 8.4163 | 0.001492629 | Uncharacterized |
| 25 | CAALFM_C405830WA | 8.3518 | 3.67121E-05 | Uncharacterized |

*Table 2 continued*

**Metal Transporters**

| 26 | CAALFM_C304210WA | 8.1065 | 3.03712E-07 | Uncharacterized |
|---|---|---|---|---|
| 27 | CAALFM_C406150CA | 7.9704 | 0.014291503 | Uncharacterized |
| 28 | CAALFM_C303370CA | 7.9704 | 1.1929E-135 | Uncharacterized |
| 29 | CAALFM_C204450WA | 7.8462 | 6.43154E-11 | Uncharacterized |
| 30 | CAALFM_C202180WA (ZRT3) | 7.7272 | 3.31888E-71 | Uncharacterized |
| 31 | CAALFM_C100270WA | 7.1398 | 4.2578E-73 | Uncharacterized |
| 32 | CAALFM_C401860CA | 7.0743 | 1.14226E-09 | Uncharacterized |
| 32 | CAALFM_C302360CA | 6.7969 | 5.99145E-37 | Uncharacterized |
| 34 | CAALFM_C600980CA | 6.6896 | 7.44423E-52 | Uncharacterized |
| 35 | CAALFM_CR06310WA | 6.6535 | 0.00233344 | Uncharacterized |
| 36 | CAALFM_C202750CA | 6.5793 | 0.000307212 | Uncharacterized |
| 37 | CAALFM_C200760CA | 6.4769 | 1.48506E-85 | Uncharacterized |
| 38 | CAALFM_C104690CA | 6.2900 | 1.03644E-05 | Uncharacterized |
| 39 | CAALFM_C304350CA | 6.2289 | 1.2123E-25 | Uncharacterized |
| 40 | CAALFM_C207620WA | 6.1661 | 6.18529E-06 | Uncharacterized |
| 41 | CAALFM_CR01910CA | 6.0167 | 0.000328533 | Uncharacterized |
| 42 | CAALFM_C702280WA | 6.0140 | 0.037653707 | Uncharacterized |
| 43 | CAALFM_C307070CA | 5.9226 | 2.70125E-54 | Uncharacterized |
| 44 | CAALFM_C405900CA | 5.9168 | 1.1249E-116 | Uncharacterized |
| 45 | CAALFM_CR07260CA | 5.8779 | 0.013967914 | Uncharacterized |
| 46 | CAALFM_C106620CA | 5.8775 | 0.016118237 | Uncharacterized |
| 47 | CAALFM_C109300CA | 5.8320 | 2.75478E-11 | Uncharacterized |
| 48 | CAALFM_CR09530CA | 5.7885 | 2.74225E-78 | Uncharacterized |
| 49 | CAALFM_C209850CA | 5.7740 | 3.08818E-07 | Uncharacterized |
| 50 | CAALFM_C601940WA | 5.5926 | 0.006011195 | Uncharacterized |
| 51 | CAALFM_C100310WA | 5.5729 | 6.50774E-57 | Uncharacterized |
| 52 | CAALFM_C601810WA | 5.5637 | 1.97617E-12 | Uncharacterized |
| 52 | CAALFM_C110060CA | 5.4134 | 4.01975E-08 | Uncharacterized |
| 54 | CAALFM_CR07300WA | 5.2272 | 7.10193E-05 | Uncharacterized |
| 55 | CAALFM_C402330CA | 5.2137 | 6.40082E-05 | Uncharacterized |
| 56 | CAALFM_C603240WA | 5.2105 | 9.75776E-08 | Uncharacterized |
| 57 | CAALFM_C305840WA (FMO1) | 5.1777 | 0.000145963 | Uncharacterized |

products mostly play a role in metal transport and utilization and in pathogenesis (or survival in the host) occurred in response to EDTA. This could indicate a compensatory mechanism being utilized by *C. albicans* in an attempt to overcome essential metal scarcity due to EDTA treatment.

## Treatment with EDTA results mostly in downregulation of genes associated with ribosome biogenesis and One-carbon metabolism in *C. albicans*

Similar to upregulated genes analyses, the top 100 downregulated genes those expression reduced to 3–100 folds in response to EDTA treatment (*Table 3*, *Figure 2D*), and 25 uncharacterized genes with unknown functions emerged. Seven of these genes CAALFM_C404230WA (orf19.9003 may encode an MFS domain-containing membrane protein), CAALFM_C112940CA (orf19.4923.1 may encode a rod shape-determining protein MreC), CAALFM_CR09350CA (orf19.7330 may encode TENA_THI-4 domain-containing protein), CAALFM_CR06430WA (orf19.8344 may encode a mediator of RNA polymerase II transcription subunit 7), CAALFM_CR08830WA (orf19.7279.1 may encode a mitochondrial ATP synthase-coupling factor 6), CAALFM_C106870CA (orf19.6222.1 may encode a thyroglobulin type-1 domain-containing protein) and CAALFM_C400530CA (orf19.11666 may encode a SurE domain-containing protein) exhibited minimum of fivefolds reduced expression in CAET (*Supplementary file 5*). We observed downregulation of 44 genes having roles in metabolic processes for example *PHO100* (Putative inducible acid phosphatase), *PLB1* (Phospholipase B), *THI4* (Thiamine biosynthetic enzyme precursor), *THI13* (Thiamin pyrimidine synthase), *GIT1* (Glycerophosphoinositol permease), *PUT1* (proline oxidase), *INO1* (inositol 1-phosphate synthase), *PUT2* (delta-1-pyrroline-5-carboxylate dehydrogenase), *LYS22* (Homocitrate synthase), *ACO2* (aconitate hydratase 2), *SNZ1* (pyridoxine (vitamin B6) biosynthetic enzyme), *BTA1*(betaine lipid synthase), *LYS4* (Homoaconitase), etc., 13 genes having roles in ribosome biogenesis and function such as *RDN18*, *RBT7*, *RPS42*, *RPS28B*, etc; and 7 genes encoding for cell wall/ membrane-associated proteins (*HSP30*, *TRY6*, *PGA10*, *HGT2*, *OPT1*, *OPT3*, *HXT5*).

The GO annotation analyses of all 388 downregulated genes revealed a total of 99 GO terms (64 biological processes, 12 molecular functions, and 23 cellular components) enriched upon EDTA treatment (*Supplementary file 2B*, *Figure 2F*). The highly significant downregulated biological processes GO terms are cellular and metabolic processes mostly involved in protein synthesis, ribosome biogenesis, and amino acid biosynthesis (GO:0009987, GO:0008152, GO:0006518, GO:0006412 etc.). Similarly, GO terms of molecular function (GO:1901363, GO:0003735, GO:0003676, etc.) genes encode for heterocyclic compounds binding, ribosome structural components, and RNA binding proteins; and GO terms of cellular component (GO:0110165, GO:0005737, GO:0005739, etc.) mostly indicate downregulated genes belong to cytoplasmic and mitochondrial components. To find out the gene networks that were inhibited upon EDTA treatment, STRING analysis was carried out (*Supplementary file 3B*). High confidence (0.7 confidence level) analysis revealed the downregulation of network proteins mostly involved in cytosolic and mitochondrial ribosomal proteins and one-carbon metabolism (*Figure 2I and J* and *Figure 2—figure supplement 1C iii*). The cytoplasmic gene network involved in protein translation showed strong nodes between 74 genes, out of which 30, 31, and 13 DEGs belonged to 40 S, 60 S, and other related functions of translation (*Supplementary file 3C*, *Figure 2I*, *Figure 2—figure supplement 1C i and ii*). Similarly, a network of 56 genes involved in the mitochondrial ribosome and translation components was observed to be suppressed (*Figure 2J*). Interestingly, ribosome structure and stability require a high concentration of divalent metals. Another network of 34 down-regulated genes involved in one-carbon metabolism was detected (*LYS2*, *GCV2*, *ARG5*, *ARO8*, *CHA1*, *ARO3*, *ASN1*, *LEU42*, *LYS9*, *HIS4*, *AAT22*, *ILV6*, *TRP5*, *GDH3*, *PUT1*, *ILV5*, *GCV1*, *ARG1*, *ARO2*, *ARO4*, *MET6*, *SAH1*, *ARG4*, *MIS11*, *LYS22*, *SER2*, *DFR1*, *LYS4*, *LYS12*, *LEU4*, *ECM42*, *ILV3*, *PUT2*, and *GCV3*) and most of these enzymes also require divalent metals as cofactor for their activities (*Ducker and Rabinowitz, 2017*; *Figure 2—figure supplement 1C iii*). One-carbon (1 C) metabolism is critical to support cellular processes like purines and thymidine biosynthesis, amino acid homeostasis (glycine, serine, and methionine), etc. Since the functions of ribosomes and 1 C metabolic enzymes require various essential metals (*Akanuma, 2021*), due to scarcity of those upon EDTA treatment, cells might have evolved a strategy to lower the ribosomal activity and metabolism by reducing ribosome and polysome composition both in cytosol and mitochondria and down-regulating

**Table 3.** List of top 100 downregulated genes in EDTA treated *C. albicans* cell (CAET).

Metabolic pathways

| No. of DEGs | Name (44 DEGs) | Fold Change | FDR p-value | Function |
|---|---|---|---|---|
| 1 | PHO100 | 100.3586958 | 1.7869E-210 | encode an enzyme phosphomonoesterase |
| 2 | PLB1 | 75.16562361 | 3.7897E-221 | Virulence; Phospholipase activity; Lipid metabolism |
| 3 | THI4 | 48.25803524 | 2.1564E-130 | Involved in thiamine biosynthesis |
| 4 | THI13 | 25.45919657 | 8.2924E-175 | Regulate IL-10 and IL-12 production |
| 5 | GIT1 | 22.7136005 | 3.2311E-112 | functions as proton symporter |
| 6 | HSP31 | 18.09639 | 2.39778E-23 | Involved in diauxic shift reprogramming |
| 7 | PHO84 | 17.26103731 | 4.3071E-163 | High-affinity inorganic phosphate/H+symporter |
| 8 | PUT1 | 14.48749801 | 7.5039E-99 | Involved in amino acid metabolism |
| 9 | INO1 | 10.74384531 | 1.5532E-128 | Encodes inositol-1-phosphate synthase |
| 10 | OPT1 | 10.46218378 | 1.83956E-90 | Involved in peptide transportation |
| 11 | PUT2 | 8.132948215 | 1.989E-99 | Amino acid metabolism regulator |
| 12 | LYS22 | 7.895086763 | 2.77628E-95 | Homocitrate synthase activity and lysine auxotrophy |
| 13 | ACO2 | 7.333020332 | 2.21018E-91 | Putative aconitate hydratase 2 |
| 14 | SNZ1 | 7.225024726 | 2.98178E-78 | Involved in pyridoxine (vitamin B6) synthesis |
| 15 | BTA1 | 7.201443893 | 2.2825E-11 | Encodes the betaine lipid synthase |
| 16 | LYS4 | 6.846020558 | 1.77724E-70 | Involve amino acid metabolism |
| 17 | CAN2 | 6.021560973 | 3.95846E-54 | Import arginine |
| 18 | FCY24 | 5.986276905 | 4.80688E-58 | Recruit vitamin B6 transport |
| 19 | THI6 | 5.891998871 | 3.5842E-104 | Thiamine biosynthesis |
| 20 | LEU1 | 5.775957021 | 5.4161E-99 | Catalyzes leucine biosynthesis pathway |
| 21 | GIS2 | 5.726657785 | 2.59522E-83 | Encodes the homologue of mammalian CNBP |
| 22 | LYS12 | 5.682928227 | 1.2519E-72 | Involved in dehydrogenation of homoisocitrate |
| 23 | CHA1 | 5.516083404 | 4.36052E-27 | Involved in nutrient acquisition/metabolism |
| 24 | FET99 | 5.171473494 | 4.85608E-18 | Involved in p-phenylenediamine oxidase; Iron transporter |
| 25 | GCV2 | 5.124244872 | 8.8508E-108 | Catabolises glycine; Induction in elevated CO2 |
| 26 | GIT3 | 4.705321982 | 1.43209E-52 | Transport glycerophosphodiester metabolites into cells |
| 27 | CYC1 | 4.533120319 | 7.41108E-12 | Encode cytochrome c |
| 28 | HXT5 | 4.370444946 | 3.87863E-30 | Uptake fructose, glucose and mannose |
| 29 | ILV3 | 4.326855778 | 2.72309E-49 | Amino acid metabolism regulator |
| 30 | OPT3 | 4.085475749 | 1.8761E-38 | Transport of sulfur-containing compounds |
| 31 | GAL1 | 4.015748921 | 6.75558E-71 | Executes metabolic functions in carbon source uptake |
| 32 | GCY1 | 3.986027968 | 4.12996E-66 | Confers hypersensitivity to toxic ergosterol analog |
| 33 | GAL10 | 3.818893745 | 1.55917E-65 | Induce biofilm; Utilize the source of galactose |
| 34 | GDH2 | 3.769049401 | 8.49692E-47 | Catalyzes deamination of glutamate to alpha-ketoglutarate |
| 35 | GCV1 | 3.737254451 | 2.09928E-39 | Catabolises glycine; Induction in elevated CO2 |

*Table 3 continued on next page*

*Table 3 continued*

Metabolic pathways

| | | | | |
|---|---|---|---|---|
| 36 | DFR1 | 3.498716699 | 3.23053E-05 | Catalyses of 7,8-dihydrofolate to 5,4,7,8-tetrahydrofolate |
| 37 | GDH3 | 3.475852592 | 9.07648E-30 | Encode NADP+-dependent glutamate dehydrogenases |
| 38 | CTP1 | 3.456185992 | 2.97824E-16 | Involved in transportion of citrate |
| 39 | TPO4 | 3.351032169 | 4.97048E-29 | Bcr1- associated repression in RPMI a/alpha biofilms |
| 40 | OPI3 | 3.327563871 | 1.89288E-15 | Biosynthesis of phosphatidylcholine |
| 41 | EGD2 | 3.290389697 | 2.54237E-30 | GlcNAc-induced protein |
| 42 | LYS2 | 3.228201144 | 3.15144E-40 | lysine biosynthesis; biofilm induced |
| 43 | NUP | 3.137825901 | 7.10193E-05 | Involved in transportation of purine nucleosides and thymidine |
| 44 | ARO3 | 3.04435509 | 1.66463E-45 | aromatic amino acid synthesis; |

Cell-wall associated

| | Name (7 DEGs) | Fold Change | FDR p-value | Function |
|---|---|---|---|---|
| 1 | HGT2 | 22.87250932 | 1.9154E-170 | Encode cell- wall associated proteins |
| 2 | PGA10 | 9.817718799 | 2.80343E-96 | Involved in Iron acquisition |
| 3 | TRY6 | 8.936709002 | 5.07311E-24 | Transcriptional regulator in biofilm |
| 4 | HSP30 | 6.401066175 | 8.86109E-17 | Stress-protective function on plasma membrane |
| 5 | ECM331 | 3.900822094 | 5.76083E-24 | Involved in cell wall biogenesis |
| 6 | ACS2 | 3.097921844 | 3.19601E-24 | antigenic during human and murine infection |
| 7 | STB3 | 3.038270294 | 5.50968E-41 | caspofungin induced |

Ribosomal

| | Name (13 DEGs) | Fold Change | FDR p-value | Function |
|---|---|---|---|---|
| 1 | RPS28B | 5.090570767 | 1.40419E-10 | Autoregulates the decapping of its own mRNA machinery |
| 2 | RPS42 | 4.347730989 | 1.02472E-32 | Enhance tolerance to fluconazole |
| 3 | RDN18 | 4.047975479 | 1.9867E-13 | component of the small (40 S) ribosomal subunit; |
| 4 | RPS12 | 3.944266266 | 1.51774E-29 | pre-rRNA processing and polysome content |
| 5 | RPP1B | 3.606949912 | 2.33237E-25 | Involved in regulation of translation elongation |
| 6 | RPS21B | 3.588439098 | 1.10333E-29 | Regulated by Nrg1, Tup1 |
| 7 | RPS13 | 3.497233295 | 3.88662E-35 | _ |
| 8 | ASC1 | 3.385291153 | 3.46635E-25 | Required for virulence in mice |
| 9 | RPL18 | 3.234852275 | 1.52941E-28 | repressed upon phagocytosis by murine macrophage |
| 10 | RPL9B | 3.193159155 | 3.27531E-29 | repressed upon phagocytosis by murine macrophages |
| 11 | RPL5 | 3.102732401 | 2.25382E-30 | repressed upon phagocytosis by murine macrophages |
| 12 | RPP2B | 3.083956604 | 5.93521E-16 | possibly involved in regulation of translation elongation |
| 13 | RPS27 | 3.03019245 | 2.15647E-21 | repressed upon phagocytosis by murine macrophage |

Others

| | Name (11 DEGs) | Fold Change | FDR p-value | Function |
|---|---|---|---|---|

*Table 3 continued on next page*

*Table 3 continued*

Metabolic pathways

| | | | | |
|---|---|---|---|---|
| 1 | PGA45 | 5.590917256 | 6.40747E-43 | Putative GPI-anchored protein of unknown function |
| 2 | RBT7 | 4.226949465 | 1.5772E-09 | Encode secreted RNase T2 |
| 3 | PEX4 | 3.972508718 | 1.24715E-12 | Spider biofilm induction |
| 4 | RME1 | 3.697681605 | 7.1604E-68 | Development of fluconazole resistance |
| 5 | THI20 | 3.271494397 | 5.58989E-53 | Spider biofilm induced |
| 6 | ASM3 | 3.235903328 | 1.59858E-31 | Possible Kex2 substrate |
| 7 | BFR1 | 3.226066337 | 1.87417E-16 | Protein involved in the maintenance of normal ploidy |
| 8 | TUF1 | 3.224254282 | 7.29461E-36 | Encodes GTPase mitochondrial elongation factor Tu |
| 9 | GAL7 | 3.207467422 | 7.9819E-49 | downregulated by hypoxia |
| 10 | SSP96 | 3.203024284 | 1.12259E-06 | F-12/CO2 early biofilm induced |
| 11 | CYB5 | 3.179133812 | 1.7903E-06 | induced in high iron |

Uncharacterized

| | Name (25 DEGs) | Fold Change | FDR p-value | Function |
|---|---|---|---|---|
| 1 | CAALFM_C400530CA | 25.251715 | 3.77714E-07 | Uncharacterized |
| 2 | CAALFM_CR08830WA | 13.160107 | 3.7788E-05 | Uncharacterized |
| 3 | CAALFM_CR06430WA | 6.746508 | 0.002469228 | Uncharacterized |
| 4 | CAALFM_CR09350CA | 6.189599 | 1.40099E-55 | Uncharacterized |
| 5 | CAALFM_C112940CA | 6.061342 | 0.037846274 | Uncharacterized |
| 6 | CAALFM_C404230WA | 5.563465 | 2.28386E-82 | Uncharacterized |
| 7 | CAALFM_C106870CA | 5.0202 | 0.017180492 | Uncharacterized |
| 8 | CAALFM_C306240CA (MRPL39) | 4.970367 | 0.002579226 | Uncharacterized |
| 9 | CAALFM_C500130CA (YML34) | 4.786879 | 5.51407E-07 | Uncharacterized |
| 10 | CAALFM_C204770WA | 4.55791 | 0.001066926 | Uncharacterized |
| 11 | CAALFM_C103620CA | 4.161708 | 4.50987E-09 | Uncharacterized |
| 12 | CAALFM_C501540WA (RPS11A) | 3.909799 | 3.0519E-17 | Uncharacterized |
| 13 | CAALFM_C503290CA (MRP10) | 3.907568 | 1.74642E-10 | Uncharacterized |
| 14 | CAALFM_C503480CA | 3.871642 | 1.41077E-32 | Uncharacterized |
| 15 | CAALFM_C208180CA | 3.782614 | 0.001381795 | Uncharacterized |
| 16 | CAALFM_C400990WA | 3.6619335 | 0.000850431 | Uncharacterized |
| 17 | CAALFM_C403500CA | 3.65632 | 0.012820924 | Uncharacterized |
| 18 | CAALFM_C204110WA | 3.438488 | 1.6049E-10 | Uncharacterized |
| 19 | CAALFM_C703560WA | 3.37104 | 3.35062E-18 | Uncharacterized |
| 20 | CAALFM_C108770WA | 3.243523 | 1.51784E-29 | Uncharacterized |
| 21 | CAALFM_CR03110WA (MSE1) | 3.172182 | 6.38954E-14 | Uncharacterized |
| 22 | CAALFM_CR03470WA | 3.162178 | 2.39698E-48 | Uncharacterized |
| 23 | CAALFM_CR08480CA (RPS29A) | 3.142855 | 2.18259E-06 | Uncharacterized |
| 24 | CAALFM_CR02380CA | 3.093999 | 0.000634778 | Uncharacterized |
| 25 | CAALFM_C400230WA | 3.0457946 | 8.36614E-08 | Uncharacterized |

essential metabolic enzymes, respectively. This might have caused a delay in cell division process while the cells maintain to survive in response to EDTA.

## EDTA alters the cell wall structure and composition

Since the cell wall and cell membrane components were upregulated and cytosolic biological processes were suppressed upon EDTA treatment, we examined any effect of EDTA on the ultrastructure of *C. albicans* cells by transmission electron microscopy. While the untreated *C. albicans* cells were oval-shaped, the EDTA-treated cells were mostly round-shaped with thicker cell walls (*Figure 3A*). The thickness of the cell wall of EDTA-treated *C. albicans* cells was estimated to be 435.023±22.3 nm (~400–600 nm) which is ~fourfolds more than that of untreated (113±31 nm, range of ~70–140 nm). To rationalize the elevated thickness of the cell wall in CAET, the composition of the fungal cell wall was verified. The cell wall of *C. albicans* is composed of an inner layer of chitin, a middle layer of β-glucans, and an outermost layer of mannan (*Kapteyn et al., 2000*). These cell wall components were estimated by staining the cells with Calcofluor white (CFW), aniline blue, and Concanavalin A tetramethylrhodamine dyes, respectively, and analyzed further by flow cytometry (*Figure 3B–D*). Our FACS analyses suggested that these components were increased by ~0.5–2.5 folds when the *C. albicans* cells were exposed to EDTA, amongst mannan was the most deposited. The perfectly round shape structure of EDTA-treated *C. albicans* cells could be due to increased chitin and glucan content as they provide rigidity to the fungal cell wall and could help cells to withstand the metal scarcity associated stress.

## Reduced level of total ribosomes in EDTA-treated *C. albicans* cells

Metal ions stabilize the secondary structure of rRNA, the association of ribosomal proteins to the ribosome, and the assembly of ribosomal subunits. Thereby, they are essential for maintaining the overall structure and translation activity of the ribosome (*Akanuma, 2021*; *Klein et al., 2004*). On the other hand, ribosomes are equally important to maintain the cellular homeostasis of metal ions. Since we found a reduction in the RNA content of ribosomal and protein metabolism components upon treatment with EDTA, we intended to estimate the total ribosomes of *C. albicans* without and with 250 μM EDTA treatment. An equal amount of exponentially grown cells of two different concentrations were processed to isolate ribosomes by sucrose density gradient centrifugation and analyzed as described before (*Pospisek and Valasek, 2013*). A clear separation of all four different forms of ribosomes: 40 S, 60 S, 80 S, and polysome was observed in the fractionations of both EDTA-treated and untreated cells (*Figure 3E*). The presence of higher amount of cell-free monosomes and polysomes indicated active protein translation in the cells. Interestingly, EDTA-treated cells possessed lesser volumes of 40 S, 60 S, 80 S, and polysome fractions than those found in untreated *C. albicans* cells.

## EDTA-treated *C. albicans* cells interacted efficiently with macrophages

Innate immune cells are the first line of defense against any invading pathogens (*Netea et al., 2008*). The immune cells like macrophages and neutrophils engulf *C. albicans* cells and produce reactive chemicals to eliminate the pathogen. As a counteractive measure, the fungal cells produce several enzymes and toxins to kill the host's cells and evade the immune system. Since the cell wall is the first interacting interface between the fungal and host cells, the altered cell wall composition could alter the recognition and phagocytosis of fungal cells by macrophages. To examine and compare fungal uptake and clearance by macrophages, we co-cultured differentially stained murine macrophages (RAW 264.7 cells stained with deep red) and untreated or EDTA-treated *C. albicans* cells (stained CFSE) in a 1:1 ratio and cells were taken out at different time points (1–3 hr) for flow cytometry analysis. A gating strategy to separate unstained and deep red stained cells during fluorescence acquisition in a flow cytometer and their analyses are depicted (*Figure 4A i and ii* and *Figure 2—figure supplement 1D*). Our FACS analysis of co-cultured cells revealed that within 1–3 hr of co-culturing, about ~78–90% of the macrophages were double-positive suggesting efficient uptake of fungal cells by the macrophages, which is equivalent to earlier reported phagocytosis efficiency (*Kumari et al., 2023*). However, a comparatively lower percentage of RAW cells was found to be interacting with CAET than untreated *C. albicans* cells (~60% to 90%). At an initial time point of incubation (1 hr), macrophages engulfed untreated and treated fungal cells with equal efficiency (~90%); however, phagocytosis reduced significantly at 3 hr in the case of CAET than *Ca* (~78 to 61%). To further

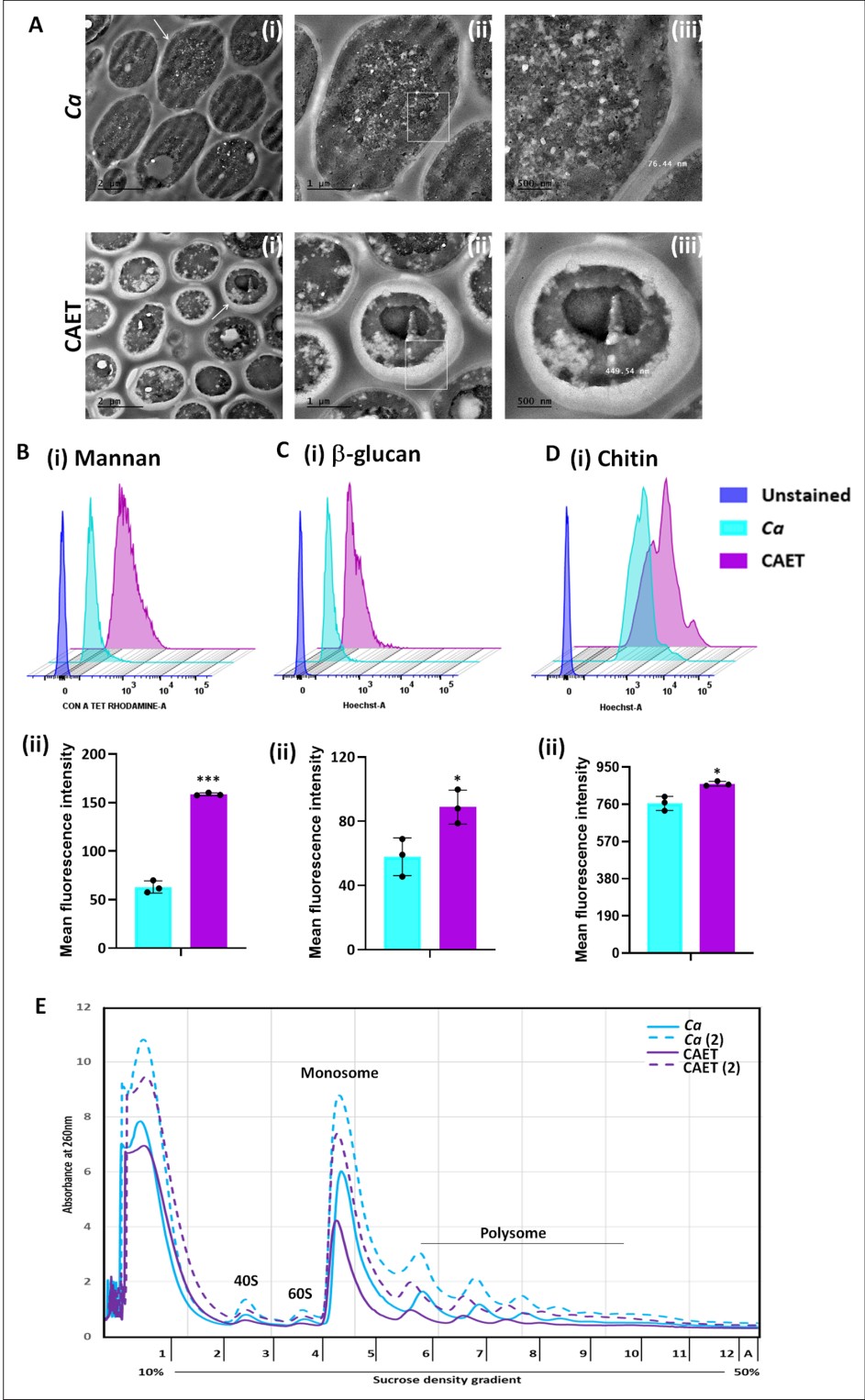

**Figure 3.** ffect of EDTA on the cell wall and polysome of *C. albicans*. (**A**) Showing TEM images of ultrathin sections of untreated (*Ca*) and EDTA-treated (CAET) *C. albicans* at different resolution. Samples visualized at ×14,000 magnification, arrow indicates the marked cell, and box indicates the focus area where thickness was measured. Scale bar = 2 μm for intact cells (**i**); scale bar = 1 μm for a single cell (ii); scale bar = 500 nm for zoomed in image of an individual cell wall (iii). (**B**) *C. albicans* cells were stained with Con A and analyzed by flow cytometry (**i**) and mean fluorescence intensity was measured to estimate the mannan level (ii). (**C**) *C. albicans* cells were stained with aniline blue and analyzed by flow cytometry (**i**) and mean fluorescence intensity was measured to estimate the

*Figure 3 continued on next page*

*Figure 3 continued*

β-glucan level (ii). (**D**) *C. albicans* cells were stained with calcofluor white and analyzed by flow cytometry (**i**) and mean fluorescence intensity was measured to estimate the chitin level (ii). Mean average of three independent biological replicates with error bars is shown. p values *<0.05 and ***<0.001 as determined by unpaired t-test. (**E**) Cell-free total ribosomes were isolated by taking two different concentrations (pellet size of 200 µL and 400 µL) of *C. albicans* cells (*Ca* and CAET) and fractionated using sucrose gradient centrifugation. Fractions were analysed and plotted. The position and transition of ribosome subunits, monosome and polysome peaks were as shown.

The online version of this article includes the following source data for figure 3:

**Source data 1.** Ultrastructure of *C. albicans* cells upon EDTA treatment and estimation of cell wall thickness, various components of cell wall, and ribosome levels.

validate it, internalized fungal cells were retrieved from the macrophages, and their survival was estimated by counting colony forming units (*Figure 4B i and ii*). The CFU analysis revealed more killing of CAET cells than *Ca* by macrophages upon longer co-incubation. This result indicated a faster clearance of metals-deprived condition grown *C. albicans* cells by macrophages. Similarly, PI staining of the co-cultured cells further confirmed more killing of macrophages when they interacted with unexposed than EDTA-exposed fungal cells (*Figure 4C i and ii*). These results implemented that *C. albicans* cells with thickened cell walls due to EDTA exposure get recognized and eliminated faster by immune cells than the naive wild type *C. albicans* cells in vitro.

## EDTA-treated *C. albicans* cells exhibit attenuated virulence in mice model of systemic candidiasis and upon immunization, they efficiently protect lethal challenge of pathogenic *C. albicans*

The systemic challenge of *C. albicans* cells induces multi-organ failure in mice similar to in humans that eventually kills the host (*Patel et al., 2023*; *Peroumal et al., 2019*). Since we observed a differential virulence of fungal cells when they were co-cultured with immune cells, we determined the disease-causing ability of these fungal cells in vivo. BALB/c mice (n=6) were injected with $5 \times 10^5$ fungal cells per mouse via the lateral tail vein and monitored their survival for a month (*Figure 5A*). Mice who suffered due to severe candidiasis were euthanized based on the humane endpoints. While the mice challenged with untreated *C. albicans* succumbed to infection within 8 days post-inoculation, the mice group injected with CAET survived similarly to the saline control group and did not show any sign of infections. This result suggested that upon EDTA treatment, *C. albicans* cell loses its virulence attributes and become non-pathogenic. Therefore, we argued whether animals immunized with CAET can protect against the pathogenic challenge of raw *C. albicans* cells. To validate, two groups of BALB/c mice (n=6) were immunized intravenously either by saline or by $5 \times 10^5$ CAET fungal cells per mouse and after 30 days, they were re-challenged with the same inoculum size of virulent form of *C. albicans* cells per mouse and their survivability was monitored. Interestingly, while the saline control sham immunized mice succumbed to fungal infections within 9 days, the CAET immunized mice were protected to the lethal re-challenge and survived till their natural death (orange *vs* brown lines). To re-validate it, the animal-challenged experiment was repeated twice on different occasions and the results were very consistent (*Figure 5—figure supplement 1*). We further confirmed the suffering of mice due to pathogenic *C. albicans* challenge by histopathology of PAS-stained kidney sections and CFU analyses (*Figure 5B and C*). The CFU analysis of the vital organs such as the kidney, liver, and spleen of killed mice groups (*Ca* and 1°Saline-2°*Ca*) confirmed the presence of heavy *C. albicans* load (*Figure 5—figure supplement 1*). The kidney of infected mice had a higher fungal load ($\sim 10^5$ cells) than the other two organs liver and spleen ($\sim 500$–$1000$ cells). PAS-stained cortex and medulla portion of the kidney depicted the presence of hyphal *C. albicans* cells. Our virulence-challenged analyses demonstrated that EDTA-treated *C. albicans* by altering its cell wall structure and gene expression reduces virulence potential drastically, and such strains might generate enough immune responses to protect pathogenic WT re-challenge. More importantly, although the effect of EDTA on *C. albicans* growth was reversible by the addition of metals in vitro, that was not the case in animals as the CAET cells were attenuated and were efficiently cleared by the host immune mechanisms as also depicted in *Figure 4*.

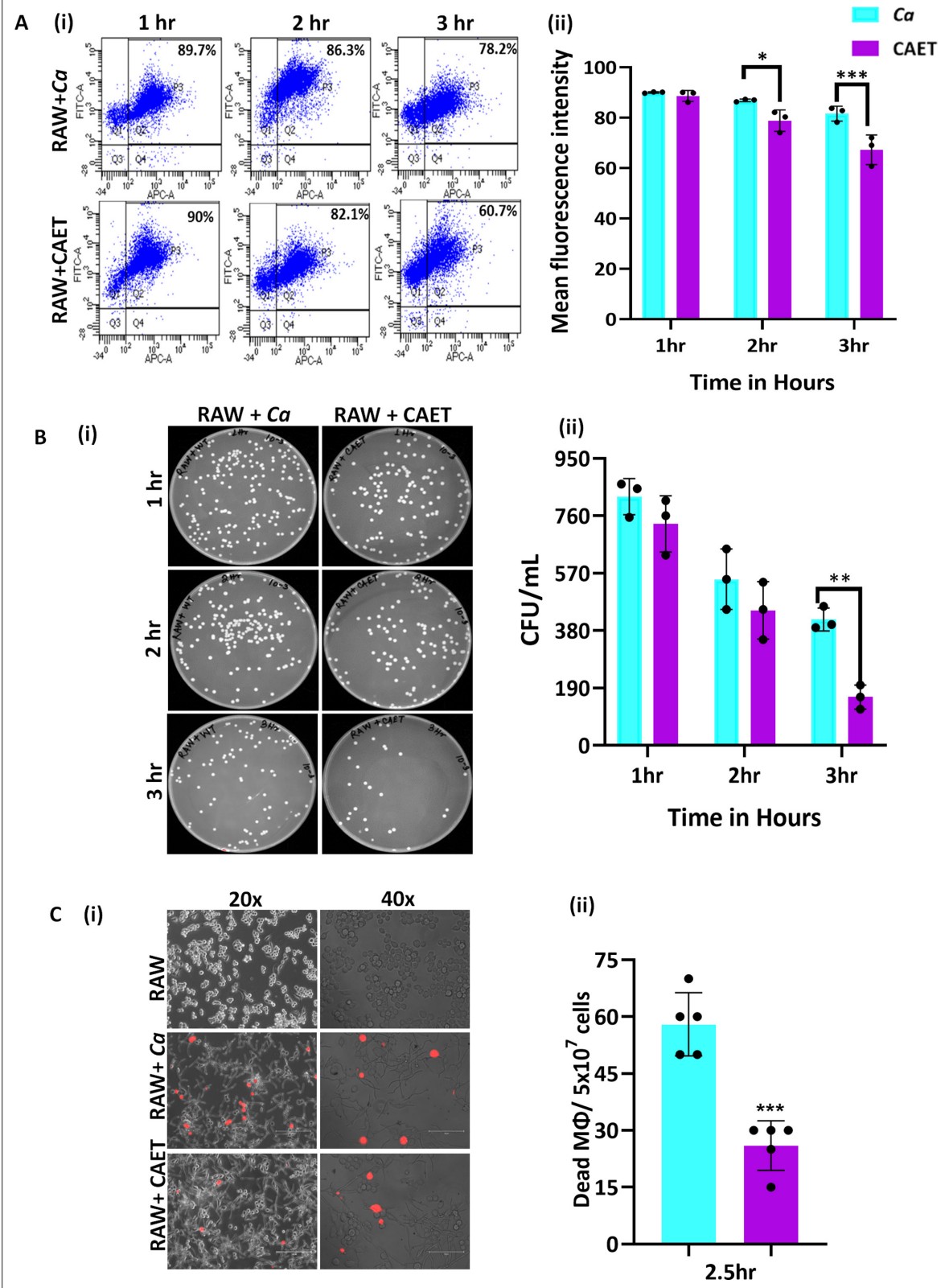

**Figure 4.** *C.albicans*-Macrophage interaction. (**A**) Deep red-stained RAW 264.7 murine macrophage cells were co-cultured with *C. albicans* (*Ca* and CAET) in 1:1 ratio. After each time point (1 hr, 2 hrs and 3 hrs) cells were pooled down and double positive cells (CFSE-FITC channel and deep red-APC channel; **Q2, P3**) were analysed by flow cytometry (**i**). Mean florescence intensity of three independent biological replicates were plotted using the GraphPad Prism software (**ii**). (**B**) From a similar co-culture experiment, *C. albicans* cells (*Ca* and CAET) were retrieved from macrophage cells and plated

*Figure 4 continued*

on YPD/Chloramphenicol plate by taking appropriate dilutions. The plates were incubated at 30 °C for 48 hr. The plates were imaged (**i**) and colony-forming unit (CFU) was determined (ii). (**C**) A similar co-culture of macrophage and *C. albicans* (*Ca* and CAET) were carried out for 2.5 hr and the dying macrophages were observed by propidium iodide staining (Red dots). Images were captured with fluorescence microscope (EVOS imaging system; Thermo Fisher Scientific) at ×20 and ×40 magnifications (**i**) and the population of dead macrophages were determined (ii). p values *<0.05, **<0.01, and ***<0.01 as determined by two-way ANOVA.

The online version of this article includes the following source data for figure 4:

**Source data 1.** Estimation of phagocytosis, fungal cells clearance and macrophage killing.

## EDTA-treated *C. albicans* cells replicate in vivo and alter circulating immune cell profiles

*C. albicans* has to replicate in the host to cause diseases. Systemic circulation of fungal cells leads to colonization in various organs; therefore, sepsis and septic shock due to bloodstream fungal infections are the major causes of morbidity and mortality (*Sahu et al., 2022*; *Spellberg et al., 2005*). Since the EDTA-treated fungal cells were nonpathogenic, it was intriguing to determine their growth and inflammation status in the host upon systemic inoculation to animals. Three groups of BALB/c mice were intravenously injected either with saline or $5\times10^5$ of untreated or EDTA-treated *C. albicans* cells per mouse. Similarly, two more vaccinated groups of mice (sham and CAET immunization) were generated and after 30 days, they were re-challenged with either saline or by WT *C. albicans*. Mice (n=5) were euthanized at various time points and various tissues were collected (*Figure 5D*). The proliferation status of fungal cells was determined by counting the fungal load in various organs by CFU analyses (*Figure 5E*). A heavy fungal load in most of the tissues except the lungs of the 3d and 7th day post-infected mice by untreated *C. albicans* was detected (cyan). As expected, the organs of the saline groups did not possess any fungal cells. While we could not detect the presence of *C. albicans* cells in the brain, liver, lungs, and spleen of animals infected with CAET in any post-days of infection, fungal cells were recovered from the kidney only in the 3rd and 7th days of post-challenge. On the 7th day, the fungal load in the kidney was ~35–40 folds less in CAET-challenged mice (~$10^4$ cells, purple bar) than in *Ca* challenged mice group (~$3 \times 10^5$ cells, cyan bar). Thus, the high fungal load on the 7th day post infection by untreated *C. albicans* cells could be the major reason for the tissue damage to cause death in animals by 9–10 days (*Figure 5A*). As we detected a gradual increase in fungal load from the 3rd to 7th days but no or undetectable level of fungal cells after 7 days of infection in mice challenged with CAET cells, it suggested that these cells replicate but get eliminated faster from the system leading to the survival of animals. Thus, CAET cells are live and divide in the host initially but as they cannot evade the immune system, they were gradually eliminated. More importantly, the kidney tissues of vaccinated re-challenged mice group (1° CAET 2° *Ca*) also possessed minimal fungal burden ($3\times10^3$ cells) only at 7th day post infection and no detectable fungal cells beyond this time point. This result suggested a faster clearance of WT cells leading to mice survival and most likely due to a robust protective immune response generated upon CAET vaccination (*Figure 5E*). PAS staining of kidneys further confirmed high fungal load in untreated than EDTA treated *C. albicans* cells in the 3rd and 7th day of post-infection. No fungal cells were detected in any of the mice groups challenged with CAET beyond 10 days of post-infection irrespective of WT re-challenge (*Figure 5—figure supplement 2A*).

Quantitative and qualitative aberrations of immune cells like neutrophils and monocytes are found to be linked to systemic candidiasis and indicate the inflammation status of the host. To get an overview of systemic levels of various immune cells upon *C. albicans* infections, the blood of different groups of mice was analyzed to determine any alterations in blood cell profile using a blood counter (*Figure 5F* and *Figure 5—figure supplement 2B*). A noticeable increase of about two- to fivefolds in peripheral WBC, monocytes, granulocytes (neutrophils, eosinophils, and basophils), and platelet levels but reduced RBC were observed in *Ca* infected mice in comparison to those in saline group mice in both 3rd and 7th days of post-infection. Interestingly, CAET infected mice showed mild alteration only in WBC and platelet counts in comparison to the control group. This result again confirms attenuated virulence of *C. albicans* upon EDTA treatment. The levels of lymphocytes, and other RBC-associated parameters (MCH, MCHC, and MCV) did not alter significantly and remained the same in all three groups of mice (*Figure 5—figure supplement 2B*).

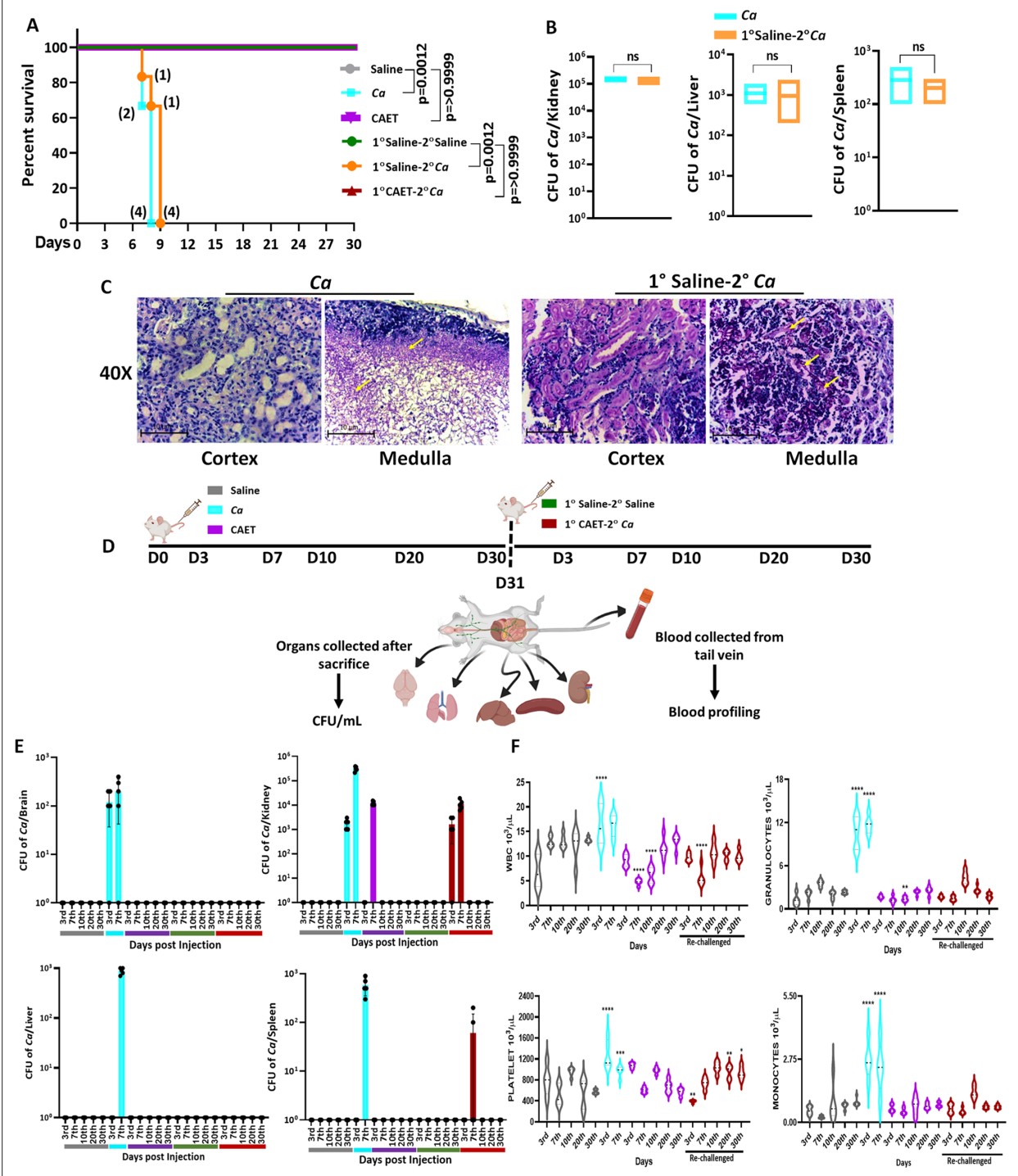

**Figure 5.** Systemic candidiasis development and progression in *C. albicans* challenged mice. (**A**) Mice (n=6/category) were injected intravenously with 5x10[5] *C. albicans* cells (*Ca*: cyan blue and CAET: purple) and saline as control (grey) and their survivability was monitored for 30 days. In a similar set of experiment, mice (n=6/category) were first immunized with CAET (represented in brown) or sham vaccinated with saline (represented in orange), and after 30 days, they were further re-challenged with *C. albicans*. Their survivability was monitored for another 30 days and a survival curve was plotted. The number of sacrificed mice were mentioned on the particular days in the survival plot. 1° and 2° suggest primary and secondary challenges, respectively. Statistical significance of the survival curve was determined using Mantel-Cox test. p values were as mentioned, otherwise they were non-significant. Y-axis was represented in Log rank scale. (**B**) Fungal load in the kidney, liver, and spleen of sacrificed mice in Log$_{10}$ scale was determined by CFU analyses. No statistical significant difference was observed between these groups. (**C**) PAS-staining for one of the kidneys carried out to visualize the fungal burden in the cortex and medulla regions under ×40 magnification. (**D**) To check the disease progression, a kinetic experiment was planned

*Figure 5 continued on next page*

*Figure 5 continued*

as indicated in the schematic diagram. A group of mice were injected intravenously with 5x10$^5$ *C. albicans* cells (*Ca*: cyan blue, CAET: purple, and 1° CAET 2°*Ca*: brown) and saline controls (grey and green) at 0 day, and five mice from each group were sacrificed on each day mentioned (3d, 7th, 10th, 20th[h], and 30[th] day). Approximately 20 µl blood was drawn from the lateral tail vein of the mice prior to sacrifice. (**E**) Fungal load in the brain, liver, kidney, and spleen in Log scale was determined by CFU analyses. (**F**) Blood parameters such as WBC, Granulocytes, Platelets, and Monocytes levels were analyzed on the above-mentioned days and graphs were plotted using the GraphPad Prism software. The statistical analyses between saline and fungal infected mice groups of same day sacrificed were carried out. p values **<0.01, **<0.01, ***<0.00, and ****<0.0001 as determined by two-way ANOVA.

The online version of this article includes the following source data and figure supplement(s) for figure 5:

**Source data 1.** Virulence and vaccine potentials of Ca Vs CAET.

**Figure supplement 1.** Systemic candidiasis development in *C.albicans* challenged mice.

**Figure supplement 2.** Kinetic analyses of fungal cells load and blood profiles of mice challenged with Ca and CAET.

## EDTA-treated *C. albicans* cells induce balanced levels of cytokines and chemokine production

Involvement of the pattern recognition receptor (PRRs) of macrophages, dendritic cells, etc. in recognizing fungal pathogens induces multiple cytokines and chemokines, which in turn activate a cascade of host immune responses. T-cells are also the major producers and effectors of these messengers. The levels of these messengers are differed in different sites of infections and responses to the virulence status of *C. albicans* (*MacCallum et al., 2009a*). For our analysis, we estimated circulating serum cytokine and chemokine levels in virulent and avirulent fungal-challenged mice and compared them with the saline group in a time-dependent manner by using a multiplex array system (*Figure 6*). Despite differential rates of fungal replication in vivo, on the 3rd day of infection, all the tested cytokines and chemokines were induced in the fungal infected mice serum irrespective of the virulence status, although with very high level in untreated *C. albicans* inoculated mice than CAET group in comparison to that in the control suggesting a prompt triggering of early inflammatory response. While we observed a gradual decline of these towards the basal levels exclusively in mice infected with CAET from 3 to 30 days post-challenge, a significant rise on the 7th day of *Ca*-challenged mice was evident. This upsurge might have created enough immunosuppressive as well as hyper-inflammatory environments allowing fungal propagation and systemic fungal infection leading to host tissue damage and mortality by the 9th day. Especially, Th-1 cytokines like IL-1β, IL-10, GM-CSF, Th-2 cytokines IL-4 and IL-13, and IL-17 from Th-17 type were highly induced on the 7th day of mice group infected with virulent form of *C. albicans*. Increased levels of anti-inflammatory cytokines like IL-2, IL-3, IL-4, IL-5, IL-9, IL-10, and IL-13 will cause immunosuppression, while the pro-inflammatory cytokines like IL-1, IL-12, IL-6, IL-17, IFN-γ, and TNF-α might lead to hyper inflammation in untreated *C. albicans* challenged mice. The levels of chemokines like RANTES, MCP-1, MIP-1 (α and β), KC, and EOTAXIN were also found to be high in the serum of fungal-infected mice, and similar observations were also found in the kidney and spleen tissues of infected murine in an earlier reported study (*MacCallum et al., 2009a*). Taken all together our data suggest that while a critical balance of the pro-and anti-inflammatory cytokines mediated immune responses is maintained for a long duration in mice challenged with CAET to induce antifungal protective immunity, both of these cytokines remained high in the pathogenic-challenged mice group leading to immunosuppression and death. Altogether this study found that CAET cells are live but non-pathogenic and boost immune responses to protect against lethal fungal challenges in a pre-clinical model; thus, CAET could be explored further as a potential whole-cell vaccine candidate.

## Discussion

This study explored the effect of metal chelators specifically EDTA as a component of nutritional immunity on *C. albicans* biology (*Figure 7*). Here we found that EDTA inhibits biofilm formation and the growth of *C. albicans* in both liquid and solid media. Other metal chelators like DTPA, aprotinin, TPEN, and CE also exhibited similar or more pronounced effects on *C. albicans*. The limited antifungal effect of EDTA on fungal colonization and growth by chelating calcium ions from the medium has been shown in earlier studies (*Gil et al., 1994*; *Pugh and Cawson, 1980*; *Sen et al., 2000*). These chelators were also shown to synergize with antifungals like azoles and caspofungin to inhibit fungal

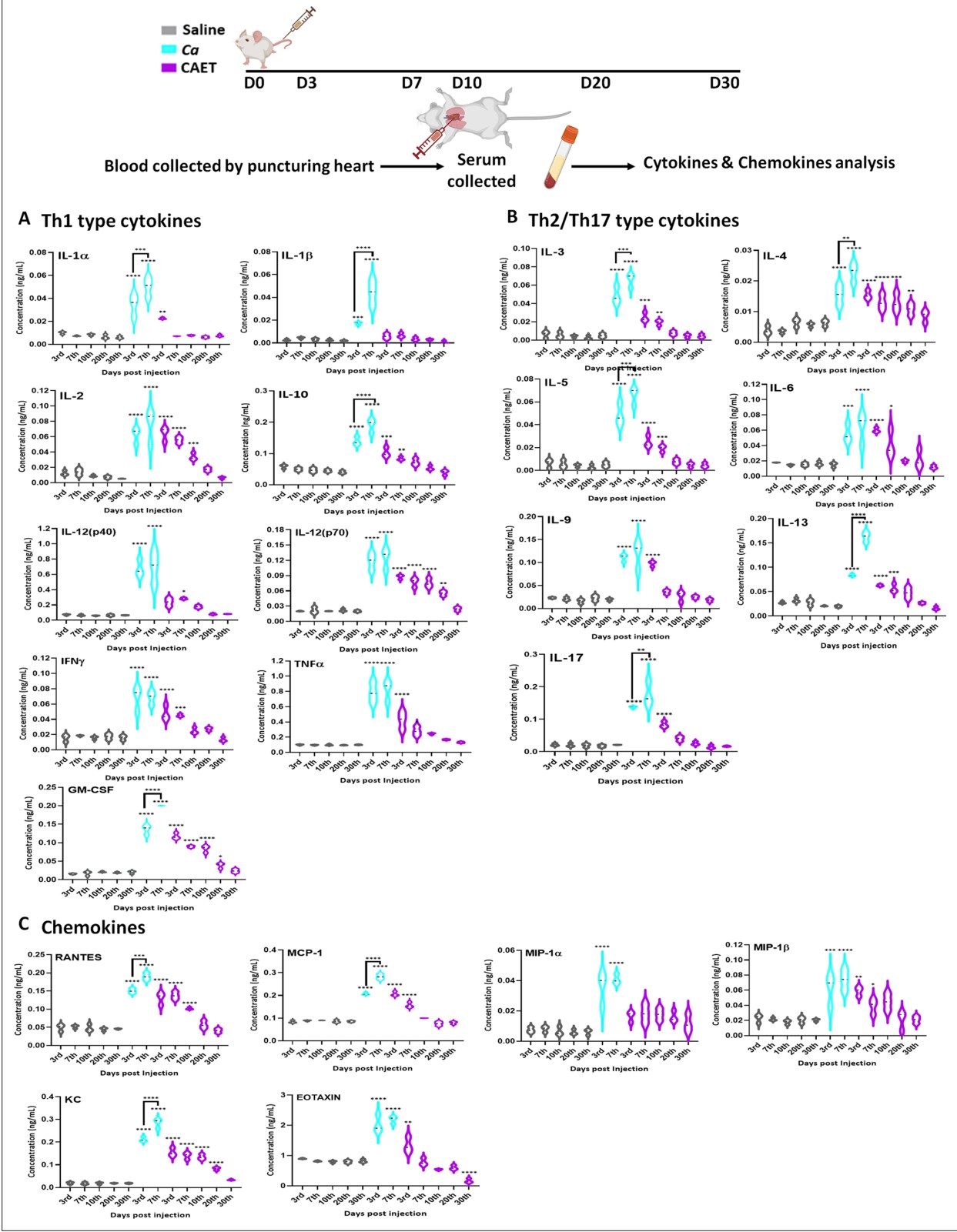

**Figure 6.** Cytokine and chemokine estimation in infected mice. The mice (n=3/category) were inoculated with *Ca* or CAET or saline and sacrificed on various days as indicated and cytokine and chemokine levels in the blood serum were quantified by using Bio-plex Pro cytokine multiplex kit. (**A**) A panel of Th-1 cytokines, that is IL-1α, IL-1β, IL2, IL-10, IL-12(p40), IL-12(p70), IFN-γ, TNF-α, and GM-CSF, (**B**) Th-2/Th-17 cytokines, that is IL-3, IL-4, IL-5, IL-6,

*Figure 6 continued on next page*

*Figure 6 continued*

IL-9, IL-13, and IL-17, and (**C**) Chemokines that is RANTES, MCP-1, MIP-1α, MIP-1β, KC, and EOTAXIN. The statistical analyses between saline and fungal infected mice groups of same day sacrificed were carried out. p values **<0.01, **<0.01, ***<0.00, and ****<0.0001 as determined by two-way ANOVA.

The online version of this article includes the following source data for figure 6:

**Source data 1.** Cytokines and chemokines estimation in fungal infected mice.

growth and biofilm formation (***Besold et al., 2018***; ***Casalinuovo et al., 2017***; ***de Oliveira et al., 2019***; ***Laskaris et al., 2016***; ***Niewerth et al., 2003***). All these reports including ours establish the importance of metals in *C. albicans* pathogenesis. Although the antifungal property of metal chelators has been known for a while, this is the first report to explore whether a critical concentration of EDTA or any other metal chelator can also attenuate the virulence of a fungal pathogen.

Next, we provide a comprehensive view of transcriptomic changes associated with the treatment of *C. albicans* with EDTA and their effect on fungal pathogenesis. Major changes in the expression of key genes and pathways associated with *C. albicans* were metal transport and homeostasis, pathogenesis processes, ribosome biogenesis, and one-carbon metabolic enzymes among others. While the cell wall and membrane-associated components were upregulated, cytosolic components were downregulated upon EDTA exposure. To our surprise, only the iron (*CFL5, CFL2, CSA2, CSA1, HAP3, FRE9, ARH2, FTR1, FTR2, FET3, SIT1*, and *FTH1*,) and zinc (*ZRT1, ZRT2, CSR1, PRA1*) transporters were upregulated upon EDTA treatment, whether these genes are also involved in the transportation of other metals like magnesium, manganese, etc. require further investigation. As reported earlier *ALR1, MRS2, LPE10*, and *MNR2* are required for magnesium transport (***Hans et al., 2019***; ***Hunsaker***

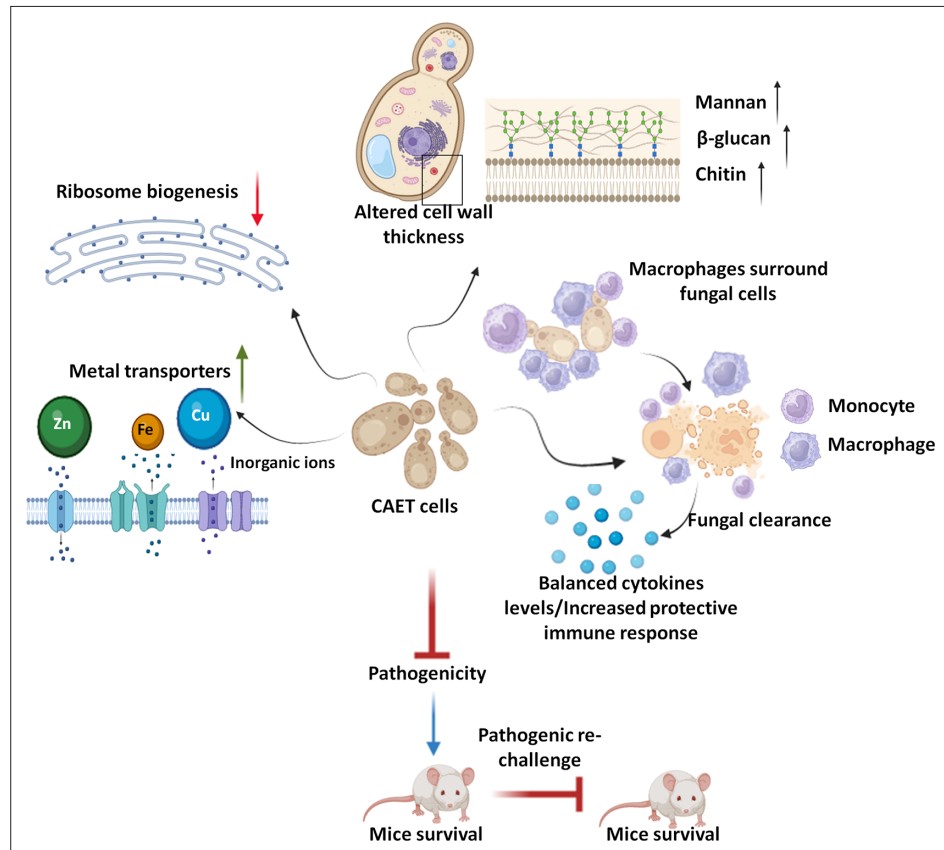

**Figure 7.** Model depicting the attributes of CAET cells. EDTA alters the cell wall thickness by altering its composition. Metal transporters and several cell wall and membrane associated genes get upregulated. To mitigate the essential metal scarcity, genes involved in ribosome biogenesis and one-carbon metabolism were down-regulated. CAET cells get phagocytosed efficiently and eliminated faster by macrophages. CAET-infected mice survived and induced robust host immune responses to protect the lethal rechallenge. Thus, CAET is a potential live whole-cell vaccine candidate.

*and Franz, 2019*; *Niewerth et al., 2003*); however, the expression of these genes remains unaltered in our analysis. Most of these upregulated genes involved in metal homeostasis are also found to be associated with fungal virulence. For example, *PRA1* and *ZRT1* collaborate to sequester zinc and mediate endothelial damage of host cells (*Citiulo et al., 2012*). Further study suggested that Pra1 blocks fungal immune recognition and inhibits recruitment, adhesion, and effector functions of both the host's innate and adaptive immune responses (*Zipfel et al., 2011*). A phosphate transporter *PHO87* was also highly upregulated upon EDTA exposure suggesting phosphate starvation, and the role of phosphate metabolism in virulence was shown in the past (*MacCallum et al., 2009b*). The outer layer of the *C. albicans* cell wall is enriched in highly glycosylated proteins like the GlycosylPhosphatidylInositol (GPI)-anchored proteins (GpiPs) (*Richard and Plaine, 2007*) and this study revealed upregulation of many such proteins (*PGA7, PGA13, PGA17, PGA19,* and *PGA31*). Some of these GpiPs are involved in cell wall synthesis, morphogenesis, and adhesion. Accordingly, these genes might have contributed to the increased thickness of the cell wall upon EDTA treatment by enhancing mannan, β-glucan, and chitin depositions. Consistent with this, an earlier report showed that deletion of *PGA13* caused increased sensitivity to Congo red, Calcofluor white, and zymolyase; delayed filamentation, increased adherence, and reduced virulence in a mouse model (*Gelis et al., 2012*). Similarly, the *pga7ΔΔ* mutant also exhibited reduced virulence in the murine model of systemic candidiasis (*Kuznets et al., 2014*). The *pga31ΔΔ* mutant shows modifications in the cell wall composition and caspofungin sensitivity (*Plaine et al., 2008*). EDTA treatment also enhanced the expression of a small network of well-characterized genes associated with filamentation and pathogenesis of *C. albicans* (*BRG1, TEC1, UME6, HWP1, HGC1, SAP5,* and *SAP6*). Secreted aspartyl proteinases (Saps) are encoded by a multigene family of at least nine members (*SAP1* to *SAP9*) and Sap4, Sap5, and Sap6 isoenzymes are important for the normal progression of systemic infection by *C. albicans* in animals (*Sanglard et al., 1997*). Hwp1 is a hyphal-specific cell wall protein and its role in virulence is debatable (*Daniels et al., 2003*; *Tsuchimori et al., 2000*). *UME6* is a transcriptional regulator that is induced in response to environmental cues and the *ume6ΔΔ* mutant exhibits a defect in hyphal extension and is attenuated for virulence in animal model (*Banerjee et al., 2008*). *HGC1* is a hypha-specific G1 cyclin-related protein involved in hyphal transition and virulence (*Zheng et al., 2004*). Transcription factors like Bcr1, Brg1, Rob1, Hgc1, and Tec1 were shown to be involved in biofilm formation as loss of any one of these regulators significantly compromised biofilm formation both in vitro and in vivo (*Lin et al., 2013*). Contrary to all these reports, despite the upregulation of these virulence genes, our study showed EDTA inhibiting biofilm formation of *C. albicans* in vitro and virulence in cellular and animal models. Among the down-regulated genes, genes of the ribosome biogenesis and one carbon metabolic process stood out. Altogether, our transcriptomics analysis suggested that due to the scarcity of essential metals upon EDTA treatment, *C. albicans* induced its metal uptake ability by expressing various transporters and increased the efficiency of the cell wall and membrane components, at the same time; it reduced the cytosolic activity to a bare minimum by down regulating ribosomal and 1 C metabolic genes to withstand metal deficiency and that likely induced a dormant-like state in *C. albicans*.

Although some of the virulence genes were upregulated upon EDTA treatment and their specific deletions are known to cause attenuated virulence in animal models, CAET cells were eliminated faster by macrophages and became avirulent in mice. It emphasizes that the virulence of *C. albicans* is very complex and multifactorial, and requires a network protein to manifest candidiasis development. In this study, downregulation of ribosome biogenesis and metabolic enzyme synthesis seem to be the predominant factor in suppressing the virulence of *C. albicans,* whereas the overexpression of cell-wall-associated virulence genes might have played a role in enhancing recognition by the immune cells and host immune responses. Unlike untreated *C. albicans* which multiplies in the host rapidly and the count reaches the maximum (>$10^5$ CFU/kidney) at day-7 in most of the essential organs causing tissue damage and death on the 9th –10th day, the CAET cells multiplied very slowly in infected organs (<$10^4$ CFU/kidney at 7th day) and were eventually cleared from the system by the 10th day. Host defense against systemic *Candida* infection depends mostly on the efficacy of phagocytosis of fungal cells by innate immune cells. This study showed an increased number of monocytes, granulocytes, and platelets in the circulation of mice infected with *Ca* than in CAET. This also suggests differential virulence properties of these two fungal strains. The fungal cell wall is the first point of contact with the innate immune system and plays an important role in recognition by neutrophils and

macrophages. β-glucan of *C. albicans* cell wall is recognized by dectin-1 /Toll-like receptor-2 (TLR2) receptor as the major PAMPs that trigger both host protective immunity and pathogenic inflammatory responses. The mannose receptor (MR) and TLR4 recognize N-linked and O-linked mannans, respectively (*Netea et al., 2015*). The thick cell wall of CAET contained a higher proportion of all the components, and these constituents probably played a key role in efficient phagocytosis and clearance by the macrophages than the untreated *C. albicans* cells.

Cytokines and chemokines play important roles in regulating inflammation and immune activation due to the presence of a pathogen. To analyse differential cytokines-mediated immune responses in infected mice, 16 cytokines and 6 chemokines from the serum were estimated. Induction of anti- and pro-inflammatory cytokines by the pathogenic *C. albicans* as early as 3–7 days post-infection might have caused immune suppression, hyper inflammation, and tissue damage promoting excessive fungal growth, sepsis, and death of the mice within 10 days. However, in CAET-infected mice, a critical balance of pro- and anti-inflammatory cytokines was developed. Both pro- and anti-inflammatory cytokines and chemokines levels gradually decreased from the 3rd day to 30th day, but a critical level above basal was still maintained to provide protective immunity, thus by immunization with CAET, animals were protected from the lethal challenge of *C. albicans*.

Differential levels of cytokine induction in untreated *Vs* EDTA-treated *C. albicans* infected mice again suggested possible involvement of T-cells as they are the major producers and effectors of these messengers. The protective immunity against fungal pathogens is mostly dependent on the balanced activation of 'protective' Th-1 and 'non-protective' Th-2 responses (*Netea et al., 2015*). Fungal infections inducing the Th-1 cytokines (e.g. IFN-γ, TNF-α, and IL-2) in the serum of infected individuals have been reported and these are involved in macrophage activation, nitric oxide production, and cytotoxic T lymphocytes proliferation to enhance phagocytosis and fungal clearance (*Netea et al., 2015*). IFNγ is central to anti-fungal resistance against systemic candidiasis as the IFNγ knockout mice are highly susceptible to disseminated *C. albicans* infection (*Balish et al., 1998*). IL-12 acts on NK and T cells to produce IFN-γ which in turn stimulates monocytes and macrophages to produce IL-12. Deficiency of IL-12 or IL-12R genes leads to enhanced fungal infections. The Th-2/Th-17 cytokines IL-4, IL-5, IL6, IL9, IL-13, and IL17 levels also increased in the serum of fungal-challenged mice. IL-4, IL-5 and IL-13 are responsible for the production of non-opsonizing antibodies, allergic reactions, and the suppression of inflammatory reactions caused by Th1 cytokines. Exogenous supplementation of recombinant IL-4 to mice increases susceptibility to invasive fungal infections, again suggesting the deleterious effects of Th2 responses (*Cenci et al., 1999*). IL-6-deficient mice are more susceptible to invasive candidiasis than wild type mice, which suggests that IL-6 release is fundamental during fungal infection and the severity of sepsis. Th17-response results in the production of IL-17, IL-21, and IL-22 cytokines. IL-17 upregulates several chemokines and matrix metalloproteases through the NF-KB and MAPK signalling pathways, leading to the recruitment of neutrophils into the sites of inflammation during fungal infections (*Haraguchi et al., 2010*; *Romani, 2004*). Not surprisingly, IL-17R null mice are highly susceptible to systemic candidiasis (*Huang et al., 2004*). Other cytokines such as TNF-α, IL-6, and GM-CSF are also involved in neutrophil recruitment during candidiasis. From the cytokine profiling, it is evident that while an optimal balance between Th-1 and Th-2/Th-17 responses seems to be maintained in CAET-challenged mice, an increased levels of these was observed in *Ca*-challenged mice. Chemokines are also critical for immune cell trafficking during pathogen attack and are released at an early stage of fungal infection. They bind to their respective receptors to trigger interleukins production to promote fungal clearance (*Traynor and Huffnagle, 2001*). This study reports higher levels of CC-type chemokines (MCP-1, KC, MIP-1α, MIP-1β, RANTES, and EOTAXIN) in infected mice and follow similar trends as cytokine in *Ca vs* CAET challenged groups. Platelets are known to produce some of these immune mediators like RANTES and PF4 with antimicrobial activity against *Candida* species, and platelet-rich plasma inhibits the growth of *Candida* (*Drago et al., 2013*).

Apart from innate and adaptive immunity, trained immunity also seems to be critical to providing immunological memory that is dependent on epigenetic reprogramming of innate immune cells (*Netea et al., 2015*). It has been suggested that the trained immunity is induced through the binding of dectin-1 to the β-glucans of *C. albicans* followed by activation of the non-canonical RAF1-cAMP signaling pathway causing epigenetic imprinting through histone methylation and acetylation, thereby, it boosts pro-inflammatory cytokines production in trained monocytes and macrophages. As the EDTA-treated *C. albicans* cells have a high level of β-glucans and upon its immunization, the

animals were protected to the lethal re-challenge of WT strain, the role of trained immunity cannot be ruled out. A detailed cellular and molecular mechanisms underlying protective immune responses in CAET challenged mice require further investigation and are awaited. Nevertheless, based on the presented results, CAET has the potential to be developed as a live whole-cell vaccine against fungal infections.

## Materials and methods

### Animal ethical clearance

Mice and protocols involving animals were approved by the Institutional Animal Ethical Committee, Institute of Life Sciences, Bhubaneswar, India, with a Permit Number ILS/IAEC-133-AH/AUG-18. The mice experiments were conducted following the guidelines of the institute.

### Animal, strains, and growth conditions

BALB/c female mice of 6–8 weeks old were procured from the institute animal house and maintained in individually ventilated cages under the standard condition with ad libitum. *C. albicans* (SC5314) strain was cultivated in YPD nutrient media. The yeast cells were stocked in 30% glycerol and kept at –80 °C for longer storage. *For all experiments, C. albicans pre-culture was made by growing cells for 16–18 hr in YPD growth media at 30 °C with continuous shaking at 200 rpm.*

### Identification of fungistatic concentration of EDTA

To determine the fungistatic concentration of EDTA, the pre-culture of *C. albicans* was diluted to an $OD_{600nm}$=0.5 ($10^7$ cells per mL) in fresh YPD media. Serial dilutions of EDTA concentration were prepared in a 96-well plate by using YPD media across multiple wells to obtain a gradient ranging from 500μM to 0.9 μM with a total volume of 150 μL in each well. Following the setup, 2 μL of the diluted *C. albicans* culture was added to each well. A positive control (represented as 0 μM) comprising of *C. albicans* culture alone and a negative control containing YPD only were also prepared. The plate was incubated at 30 °C under a static condition for 24 hr. The optical density at 600 nm of the growth was measured using a Perkin-Elmer plate reader. The experiment was performed with four technical replicates. Further, certain concentrations of EDTA treated cells post-24 hr treatment (0–250 μM) were selected from the 96-well plate and their respective dilutions were spotted on the YPD agar plate to determine the viability of the *C. albicans* cells. The plate was incubated at 30 °C for 24 hr and imaged.

### Growth curve and CFU analysis

The overnight *C. albicans* pre-culture was diluted to an $OD_{600nm}$=0.5 in YPD media, with each tube containing a final volume of 10 ml. Except one, these cultures were treated individually with 8 μM $MgSO_4$, 8 μM $ZnCl_2$, 8 μM $FeCl_2$, 8 μM $MnCl_2$, 250 μM EDTA +8 μM $MgSO_4$, 250 μM EDTA +8 μM $ZnCl_2$, 250 μM EDTA +8 μM $FeCl_2$, and 250 μM EDTA +8 μM $MnCl_2$. Similarly, the metal chelators used for the other analyses were 250 μM DTPA/pentetic acid, 250 μM BPTI/Aprotinin, 100 μM TPEN, and 100 μM CE. Over the next 24 hrs, the $OD_{600nm}$ and colony forming unit efficiency of the untreated and treated cultures were monitored. For CFU analysis, the cultures were serially diluted in autoclaved distilled water and around 100–150 μl of diluted samples were spread onto the YPD agar plates. The plates were incubated, colonies calculated, and the graph was plotted against time using Graph Pad Prism version 8.0. A similar growth curve and CFU analyses of *C. albicans* culture were also carried out in the presence of 250 μM EDTA, and 8 μM $MgSO_4$ was supplemented after 6 hr of treatment. The experiment was repeated thrice with biological duplicates.

### Estimation of cytotoxic effect of 250 μM EDTA on *C. albicans* cells using SYTOX green and PI staining

In order to compare the viability of *C. albicans* cells in untreated and EDTA treated condition, *C. albicans* cells ($OD_{600nm}$=0.5) were grown without and with 250 μM EDTA at 30 °C and 200 rpm for 12 hr. An equal volume of cells were harvested at various time intervals, and washed two times and resuspended in 1 mL 1 X PBS. SYTOX green (Thermo Fischer Scientific) staining was conducted at a final concentration of 1 μg/mL, followed by a 30 min incubation at 30 °C in the dark. Stained cells were washed and resuspended in 500 μL 1 X PBS and data was acquired in BD LSR Fortessa Flow cytometer with

excitation by a blue laser at 488 nm. Data were analyzed using FlowJo software and were exported in the JPEG format. Similarly, another set of cell pellet was stained with propidium iodide (200 ng/mL) and images were captured using a fluorescence microscope (EVOS imaging system; Thermo Fischer Scientific). About 1000 cells were counted and the % of PI stained cells were determined. The experiment was repeated thrice with biological duplicates.

## Morphology analyses

To determine the effect of 250 µM EDTA on the morphology of *C. albicans* cells, from the above grown cultures, cells were collected at various time intervals and the cellular phenotypes were observed under a 40 X Leica DM500 microscope. The percentage of singlet, doublet, pseudo-hyphae, and hyphae cells were estimated. Since serum induces germ tube formation, *C. albicans* cultures were grown in the presence of 10% serum (FBS; Gibco) and without and with 250 µM EDTA at 37 °C. Germ tube lengths were observed under the light microscope and length was measured using ImageJ software. The experiment was repeated thrice with biological duplicates.

## Biofilm formation assay

Biofilm of *C. albicans* was developed by growing diluted overnight culture in YPD broth ($OD_{600nm}$=0.5) in a 24-well polystyrene plate for 24 hr in a static condition at 30 °C as described before (*Bose et al., 2023*). After, 24 hr of biofilm formation, the biofilms were either kept untreated or treated with 250 µM EDTA or mentioned concentrations of other metal chelators (DTPA, Aprotinin, TPEN, and CE), 8 µM $MgSO_4$, 8 µM $ZnCl_2$, 8 µM $FeCl_2$, 8 µM $MnCl_2$, 250 µM EDTA +8 µM $MgSO_4$, 250 µM EDTA +8 µM $ZnCl_2$, 250 µM EDTA +8 µM $FeCl_2$, and 250 µM EDTA +8 µM $MnCl_2$; and again, incubated for another 24 hr. After a total 48 hr of incubation, the supernatant was gently removed without disturbing the biofilm and the wells were washed with 1 X PBS twice. The biofilm so obtained was treated with 0.1% crystal violet and left for 20 min to stain at 25 °C. The plate was allowed to air dry and the bound dye was then re-suspended in 33% glacial acetic acid, and absorbance was recorded at 570 nm using a Perkin-Elmer plate reader. The experiments were repeated twice with biological triplicates. Similarly, biofilms were formed and treated in 8 well-chambered slides, images were obtained after staining with 1% acridine orange using Leica TCS SP8 confocal scanning system with where excitation at 483 nm and emission with 500–510 nm band-pass filter were used.

## RNA extraction and sequencing

The pre-culture of *C. albicans* was diluted to an $OD_{600nm}$=0.5 ($10^7$ cells per mL) in 5 ml of YPD media. The culture was treated with 250 µM EDTA and allowed to incubate along with the untreated control culture for the next 6 hrs at 30 °C with proper agitation. The total RNA was isolated by using MagSure All-RNA Isolation Kit, according to the manufacturer's instructions. The absorbance ratios of 260/280 and 260/230 were estimated using NanoDrop2000 to assess the concentration of RNA samples. Samples were also examined in 1% agarose gel run at 70 V. The RNA was stored at –80 °C. Before RNA sequencing, QC was performed to check the quantity and integrity of RNA samples by using NanoDrop One and Qubit. The integrity of the RNA was checked by running on an Agilent Tapestation 4200 RNA HS Screentape to determine the RNA Integrity Number (RIN). For library preparation, RNA with >8 RIN values was processed. About 1 µg of RNA samples was depleted of Yeast rRNA using QIAseqFastSelect−rRNA yeast kit (Cat. no. 334215), according to the manufacturer's protocol. The NEBNext(R) Ultra II Directional RNA Library prep kit from Illumina (Cat. no. E7760L), was used to construct double-stranded cDNA libraries from the rRNA-depleted RNA. The cleaned libraries were quantitated on Qubit(R)flurometer and appropriate dilutions were loaded on a High sensitivity D1000 screen tape to determine the size range of fragments and the average library size (288–311 bp), followed by sequencing using Illumina NovaSeq.

## Transcriptomics analysis

RNA-seq using Illumina generated raw data in FASTQ format. The raw data was analyzed using CLC genomics workbench 22.02 and quality control was assessed using the *QC plugin* of CLC to generate a minimum read length of 151 base pair. In order to filter and remove rRNA reads, *SortMeRNAv4.3.6* was used. Further quality trimming was performed to remove adaptors and low-quality ambiguous bases. Mapping of trimmed reads performed using *C. albicans* SC5314 reference genome (assembly

ASM18296v3) downloaded from NCBI database. The uniquely mapped reads were selected as default followed by transcripts per million (TPM) as raw expression value. Gene expression (GE) track was used for profiling the differential gene expression, subsequently normalizing the trimmed mean of M values (TMM) using the edgeR method. In order to find the significant genes, two cut-off parameters; false discovery rate (FDR) cut-off ≤0.05 and log2 fold change (FC) cut-off ≥1 were applied to produce a differential gene expression (DEG) profile. Principal component analysis (PCA) was performed within R to project the high dimensional dataset onto a 2D or 3D plain based on the principal component specifying larger-sized variabilities. Data was represented and visualized through a heat map using the normalized log CPM values. The volcano plot was used to feature the gene of interest at the upper left- and right-hand corners, where log2 fold changes were plotted on the x-axis, and the -log10 p-values plotted on the y-axis. Pathway enrichment analysis was created based on gene ontology (GO), where GO terms were over-represented in a set of differentially expressed genes using ignore features with mean RPKM = 5, minimum absolute fold change = 1.5, and FDR $P$-value ≤0.05. KOBAS (KEGG Orthology Based Annotation System), a web server for gene/protein functional annotation (Annotation module) and functional set enrichment (Enrichment module) was used for pathway enrichment analysis. Gene interaction networks and protein-protein associations have been created using the STRING app. STRING software version 11.5 was used to undertake active parameters like text-mining, experiments, databases, co-expression, neighborhood, gene fusion, and co-occurrence for upregulated DEGs, while for downregulated DEGs, only the experimental parameter was considered. An interaction score of 0.7 (High confidence) was followed for all the STRING analyses.

## Real-time and qRT-PCR

To validate the RNA seq data, cDNA was synthesized from 1000 ng RNA using a high-capacity cDNA synthesis kit (Thermo Fischer Scientific). The cDNA stock was diluted 30 times with nuclease-free water to prepare the working cDNA. For qRT, 5 µl of diluted cDNA along with 10 pmole of each from forward-reverse primers, and 2 x SYBR Green reagent (Applied Biosystems PowerUp SYBR Green Master Mix, Thermo Fischer Scientific) was used. The amplification program consisted of the cyclic condition comprising 95 °C for 10 min on the holding stage, 95 °C for 15 s, and 60 °C for 1 min on the PCR-cyclic stage, and 95 °C for 15 sec, 60 °C for 1 min and 95 °C for 15 s on melting-curve stage, and 4 °C hold at one cycle (QuantStudio 3.0, Applied Biosystems). The mean $C_t$ value of technical replicates from the standard curve was analyzed and plotted using The GraphPad Prism 8.0. Semi-quantitative RT-PCR assay was preceded with 10 X PCR buffer, 10 mM dNTPs, 10 pmole concentration of each forward and reverse primers, and DNA Taq Polymerase. Samples were checked on agarose gel including *GAPDH* as a housekeeping gene. All the PCR primers and cyclic conditions are mentioned in *C. albicans* **Supplementary file 6**.

## Transmission electron microscopy

The ultrathin section of *C. albicans* and CAET cells were examined and visualized by TEM. The *C. albicans* cells were subjected to aldehyde for pre-fixation and osmium tetraoxide for post-fixation followed by contrast staining using uranyl acetate as described before (*Patel et al., 2023*). The hardened sample blocks were sectioned using Leica EM UC7 microtome and the samples on the copper grids were visualized under the JEM-2100Plus JEOL TEM imaging machine.

## Measurement of cell wall contents by flow cytometry

The pre-culture of *C. albicans* was diluted to an $OD_{600nm}$=0.5 in 5 mL of YPD media. The culture was then treated with 250 µM EDTA and allowed to grow along with the untreated culture for 6 hr at 30 °C and 200 rpm. For β (1, 3)-glucan estimation, aniline blue dye (Sigma, Cat. no. B8563) was added to $1\times10^6$ *C. albicans* cells/ml in a 1:1 ratio and incubated for 30 min at 30 °C. For chitin estimation, $1\times10^6$ cells/ml were treated with 2.5 µg concentration of Calcofluor White (Sigma, Cat. no. 910090) and incubated for 15 min at 25 °C. Likewise, for the estimation of mannan, we used 10 µl of concanavalin A (1 mg/ml, Thermo Fischer Scientific, Cat. no. C860). The stained cells were centrifuged at 12,000 g for 1 min. The cell pellet was washed twice with 1 ml 1 X PBS and finally resuspended in 500 µl 1 X PBS. The stained cells were then transferred to FACS tubes for acquisition. All the control and treated samples were acquired by BD LSR Fortessa Cell Analyzer using UV laser (350 nm) with a bandpass filter (450 nm/500 nm). The mean fluorescence intensity was obtained from the unstained

and stained yeast cells and processed in flowJo software version 10.8.1. The assays were repeated twice with biological triplicates.

## Polysome isolation and profiling

Polysome and its fractionation were carried out by using a protocol described before with slight modification (*Pospisek and Valasek, 2013*). Briefly, to isolate cell-free total ribosomes, *C. albicans* pre-culture was diluted to $OD_{600nm}$ of 0.5 in YPD media to obtain 250 ml each in two flasks. While one of the cultures was treated with 250 µM EDTA, the other flask was kept untreated. The sub-cultures designated as *Ca* and CAET, respectively, were allowed to grow for ~6 hr at 30 °C and 200 rpm until the $OD_{600\ nm}$ becomes 0.8–1.0 (exponential phase). Cycloheximide (100 mg/ml) was added to the cultures 5 min before harvesting the cells. The cultures were kept for 15 min on ice and then centrifuged at 3000 × *g* at 4 °C for 5 min. The pellet was washed using 0.1% autoclaved DEPC treated water and resuspended in 1.3 times total volume of the pellet weight using a lysis buffer (1 M Tris-Cl, 2 M KCl, 1 M $MgCl_2$, 10 % NP-40, 5 µl RNase Inhibitor, 20 X Protease Inhibitor EDTA free, 200 µg/mL Cyclohexamide, DEPC $H_2O$). The re-suspended pellet was mixed thoroughly and 0.6 volume of acid-washed glass beads were added and homogenized in five cycles, each comprising 40 s of vortexing at 6 m/s speed, with 2 min of ice incubation in between the cycles. The suspension was centrifuged at 3000 × *g* at 4 °C for 10 min, the top lipid layer was carefully removed, and the supernatant was collected in a fresh tube. Subsequently, two additional rounds of centrifugation were carried out at 13 K rpm at 4 °C and each for 30 min. After each centrifugation, the supernatant was collected and transferred to a fresh Eppendorf tube. The absorbance of the supernatant was checked at 260 nm and the final OD for both the samples was adjusted to 20. For the preparation of a linear sucrose gradient, 10% and 50% sucrose were taken in polyallomer centrifuge tubes and the gradient was formed using a gradient maker (Gradient Master from BioComp). The cell free extracts (200 or 400 µl of OD = 20) were added to the 10–50% sucrose density gradients carefully, and ultracentrifuged using a SW41Ti rotor and Beckman Coulter Ultracentrifuge at 39 K rpm for 2.5 hr at 4 °C with the vacuum level of 200 microns. After the centrifugation, the fractions were collected carefully and the polysome profile was analyzed using the Density Gradient Fractionation System, which continuously scanned absorbance at 254 nm. The Density Gradient Fractionation System (BIOCOMP) was switched on approximately 1 hr before fractionation to ensure its readiness.

## Macrophage interaction

About $5x10^5$ RAW 264.7 macrophage cells (ATCC) were seeded in a 12-well polystyrene plate containing DMEM supplemented with 10% FBS. After 24 hr of incubation, macrophages were washed with only DMEM and stained with a cell tracker deep red dye. Cells were routinely tested and ensured no mycoplasma contamination. After 45 min of incubation at 37 °C in a $CO_2$ incubator, the wells were washed with DMEM. The same number ($5x10^5$) freshly harvested *C. albicans* cells (*Ca* and CAET) were stained with CFSE for 1 hr. After straining, they were washed with 1 X PBS and resuspended in DMEM media. The CFSE-stained fungal cells were added to each well having deep red-stained RAW cells and co-cultured at 37 °C. The control wells contained stained RAW cells only. At three different time points 1 hr, 2 hr, and 3 hr, the supernatant was discarded from the well to remove any non-phagocytosed fungal cells. The adhered cells were scrapped and transferred into the FACS tubes. After centrifugation, the cell pellet was resuspended in 500 µl of FACS buffer. The stained RAW cells were used for making a suitable gating strategy during fluorescence acquisition in a flow cytometer. The phagocytosis was determined by gating the cells and calculating the percentage of double-positive cells (CFSE-FITC & deep red- APC). For fungal clearance assay, a similar co-culture experiment was carried out by taking unstained macrophage cells but stained *C. albicans* cells. After completion of each time interval, the supernatant was aspirated to remove any free fungal cells and 1 ml of autoclaved warm distilled water (50– 60°C) was added to each well to release the phagocytosed *C. albicans*. The cells were scrapped and pooled down to the tube. Serial dilutions of cells were plated on an YPD plate treated with chloramphenicol. The plates were incubated at 30 °C for 48 hr and CFU was determined. In macrophage killing assay, the wells comprising the macrophage-fungal cells co-culture were washed with 1 ml 1 X PBS and stained with 200 ng/mL propidium iodide (MP-Biomedicals). Images were captured under a fluorescence microscope (EVOS imaging system; Thermo Fischer Scientific) after 2.5 hr of incubation. The experiment was carried out in triplicates.

## Systemic candidiasis in mice

Six to 8 weeks BALB/c female mice were housed in a controlled and ventilated environment. To prepare the inoculum, the *C. albicans* pre-culture was diluted to an $OD_{600nm}$=0.5 ($10^7$ cells per mL) in YPD. About 10 ml of the diluted culture was grown further either untreated or treated with 250 µM EDTA for 6 hr at 30 °C with shaking at 200 rpm. Both untreated and treated cells were harvested, re-suspended in 1 X PBS, and cells were counted by using Neubauer chamber slide and online hemocytometer software (HemocyTap), and further confirmed by plating and CFU counting. After 7 days of acclimatization, depending upon the experiments, groups of mice (n=5/6) were challenged intravenously either with $5 \times 10^5$ CFU of *C. albicans Ca* or CAET or the same volume (100 µl) of saline. The survivability of mice was monitored for 30 days. Based on the humane endpoints mice were euthanized. For kinetic analysis, mice were sacrificed on mentioned days of post infection. Various essential organs (brain, lungs, kidney, liver, and spleen) were excised for CFU analysis and PAS staining. For blood cell profiling and cytokine estimation, blood was drawn just before sacrificing. CFU analysis was carried out immediately after the dissection. The organs were thoroughly washed with PBS, homogenized in 1 mL of PBS, and appropriate serial dilutions were plated on YPD agar plates supplemented with 100 µg Chloramphenicol. The plates were incubated overnight to determine CFU/organ. The CFU/organ in $Log_{10}$ scale and survival curves were plotted using GraphPad Prism 8 software. Mouse kidney paraffin sections were stained with PAS reagents as per the manufacturer's instructions and imaged under a Leica DM500 microscope.

## Blood cells profiling

Blood was obtained by slitting a small incision on the mice tail with the help of a scalpel and collected in tubes pre-filled with $K_2$-EDTA in a 1:10 ratio (Blood: $K_2$-EDTA). About 20 µl EDTA-anti coagulated blood samples were taken up in plastic capillaries (Kunstsoff-Kapillaren; 100120) and examined using an Exigo Blood Analyzer within 2–4 hr of blood collection.

## Cytokine and chemokine estimation

The BALB/c mice (n=3) from each category were intravenously injected with $5 \times 10^5$ CFU/ml *Ca* and CAET, and saline was taken as control. The mice were euthanized and the heart was punctured to obtain ~500 µL of blood. The collected blood was stored at 4 °C overnight in an inclined position. Following the day, serum was collected carefully from each blood sample and transferred to a fresh tube. The collected serum was centrifuged at 5000 g for 20 min. The supernatant was collected and the level of the cytokines and chemokines was estimated by using Bio-Plex ProTM Mouse Cytokine standard 23-Plex (Bio-Rad, USA, Cat. no. 10014905). The kit was inbuilt and supplemented with 23 analytes. The graph was plotted using the GraphPad Prism 8 software.

## Acknowledgements

We thank Mr. Sitendra Prasad Panda and Mr. Paritosh Nath for their technical assistance. Dr. Sarita Jena, Mr. Biswajit Patro, and Mr. Subrat Kumar Naik for technical assistance and nurturing of the animals were highly appreciated. Ms. Tanvi Sinha and Dr. Amaresh Panda for helping us during polysome isolation and profiling. We thank our other laboratory colleagues for their thoughtful discussion. SB and SRS are grateful for SERB/UGC-Senior Research Fellowships and AD is thankful for the DBT-RA fellowship. ILS animal house, TEM Imaging, and FACS facilities are highly acknowledged. This work was supported by the intramural core grant from ILS and extramural research funds from DBT, India (BT/PR32817/MED/29/1495/2020).

## Additional information

### Competing interests

Swagata Bose: listed as an inventor in a related patent application. 202331056818. Narottam Acharya: listed as inventor in a related patent application. 202331056818. The other authors declare that no competing interests exist.

## Funding

| Funder | Grant reference number | Author |
|---|---|---|
| Department of Biotechnology, Ministry of Science and Technology, India | BT/PR32817/ MED/29/1495/2020 | Narottam Acharya |

The funders had no role in study design, data collection and interpretation, or the decision to submit the work for publication.

## Author contributions

Swagata Bose, Resources, Software, Formal analysis, Supervision, Validation, Investigation, Visualization, Methodology, Writing - original draft, Writing - review and editing; Satya Ranjan Sahu, Abinash Dutta, Resources, Software, Investigation, Methodology, Writing - original draft, Writing - review and editing; Narottam Acharya, Conceptualization, Data curation, Supervision, Funding acquisition, Writing - original draft, Project administration, Writing - review and editing

## Author ORCIDs

Narottam Acharya (iD) http://orcid.org/0000-0001-8858-5418

## Ethics

Mice and protocols involving animals were approved by the Institutional Animal Ethical Committee, Institute of Life Sciences, Bhubaneswar, India, with a Permit Number ILS/IAEC-133-AH/AUG-18. The mice experiments were conducted following the guidelines of the institute.

Reviewer #2 (Public Review): https://doi.org/10.7554/eLife.93760.3.sa1
Reviewer #3 (Public Review): https://doi.org/10.7554/eLife.93760.3.sa2
Author response https://doi.org/10.7554/eLife.93760.3.sa3

# Additional files

## Supplementary files

• Supplementary file 1. List of top 100 upregulated genes in EDTA treated *C. albicans* cell (CAET) with detailed information.

• Supplementary file 2. GO annotation analyses of upregulated genes and downregulated genes. (**A**) GO annotation analysis for 411 upregulated genes and (**B**) 388 downregulated genes.

• Supplementary file 3. STRING cluster analyses of DEGs. (**A**) STRING cluster analysis for 411 upregulated genes, (**B**) 388 downregulated genes, and (**C**) 74 downregulated genes out of 388 DEGs specific to 40 S and 60 S ribosomal subunits.

• Supplementary file 4. STRING cluster analysis for 33 DEGs specific to pathogenesis.

• Supplementary file 5. List of top 100 downregulated genes in EDTA treated *C. albicans* cell (CAET) with detailed information.

• Supplementary file 6. Oligonucleotides and PCR conditions.

• MDAR checklist

## Data availability

Raw RNA seq reads were deposited in a public database and can be retrieved by using the accession number PRJNA1010013. The rest of the data has been provided in the article.

The following dataset was generated:

| Author(s) | Year | Dataset title | Dataset URL | Database and Identifier |
|---|---|---|---|---|
| Acharya N | 2023 | Transcriptomic analysis of various strains of *Candida albicans* | https://www.ncbi.nlm. nih.gov/bioproject/ PRJNA1010013/ | NCBI BioProject, PRJNA1010013 |

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
