## [Editor Report · eLife assessment]

This study presents a **useful** strategy in which the authors devised a simple method to attenuate *Candida albicans* and deliver a live whole-cell vaccine in a mouse model of systemic candidiasis. The reviewers are not convinced about the completeness of the study: the strength of the evidence is **incomplete** and could be augmented with additional experiments to more fully characterize vaccine efficacy and host immune responses.

---

## [Referee Report · Reviewer #2 (Public Review)]

Summary:

Invasive fungal infections are very difficult to treat with limited drug options. With the increasing concern of the drug resistance, developing antifungal vaccine is a high priority. In this study, authors studied the metal metabolism in *Candida albicans* by testing some chelators, including EDTA, to block the metal acquisition and metabolism by the fungus. Interestingly, they found EDTA treated yeast cells grew poorly in vitro and non-pathogenic in vivo in a murine model. Mice immunized by EDTA-treated Candida (CAET) were protected against challenge with wild type Candida cells. RNA-Seq analysis to survey the gene expression profile in response to EDTA treatment in vitro revealed upregulation of genes in metal homeostasis and down regulation of ribosome biogenesis. They also revealed an induction of both pro- and anti-inflammatory cytokines involved in Th1, Th2 and Th17 host immune response in response to CAET immunization. Overall, this is an interesting study with a translational potential.

Strengths:

The main strength of the report is that authors identified a potential whole cell live vaccine strain that can provide a full protection against candidiasis. Abundant data both on in vitro phenotype, gene expression profile and host immune response have been presented.

Weaknesses:

A weakness is that the immune mechanism of CAET mediated host protection remain unclear. The immune data is somewhat confusing. Authors only checked cytokines and chemokines in blood. The immune response in infected tissues and antibody response may be investigated.

Another potential concern is that using live wild type Candida cells treated with EDTA may still have chance to evolve and become infectious, considering that these treated cells still proliferate in vivo. Some of the gene regulation profiles may be transit and subjected to reverse, adding to the safety concern.

---

## [Referee Report · Reviewer #3 (Public Review)]

Summary:

The authors are trying to find a vaccine solution for invasive candidiasis.

Strengths:

The testing of the antifungal activity of EDTA on Candida is not new as many other papers have examined this effect. The novelty here is on the use of this such EDTA treated strain as a vaccine to protect against a secondary challenge with wild-type Candida.

Weaknesses:

However, data presented in Fig. 5 and in Fig. 6 are not convincing and need further experimental controls and analysis as the authors do not show a time-dependent effect on the CFU of their vaccine formulation. Specific points are below.

Methodology used is also an issue. As it stands, the impact is minor, if any.

Comments on revised version:

The data provided in the revised paper are simply not satisfactory and do not give confidence that a rigorous design and methodologies were used to obtain the results illustrated in this paper.

---

## [Author Response]

The following is the authors’ response to the original reviews.

We would like to thank both Editors and reviewers for their valuable time, careful reading, and constructive comments. The comments have been highly valuable and useful for improving the quality of our study, as well as important in guiding the direction of our present and future research. In the revised manuscript, we have incorporated the necessary changes including additional experimental data as suggested; please find our detailed pointby-point response to the reviewer’s comments and the changes we have made in the manuscript as follows.

**Reviewer #1 (Public Review):**
In this work, the authors have explored how treating *C. albicans* fungal cells with EDTA affects their growth and virulence potential. They then explore the use of EDTA-treated yeast as a whole-cell vaccine in a mouse model of systemic infection. In general, the results of the paper are unsurprising. Treating yeast cells with EDTA affects their growth and the addition of metals rescues the phenotype. Because of the significant growth defects of the cells, they don't infect mice and you see reduced virulence. Injection with these cells effectively immunises the mice, in the same way that heatkilled yeast cells would. The data is fairly sound and mostly well-presented, and the paper is easy to follow. However, I feel the data is an incremental advance at best, and the immune analysis in the paper is very basic and descriptive.Strengths:Detailed analysis of EDTA-treated yeast cellsWeaknesses:Basic immune data with little advance in knowledge.No comparison between their whole-cell vaccine and others tried in the field.The data is largely unsurprising and not novel.

Reply: Thank you so much for appreciating our effort to generate a whole cell anti-fungal vaccine by treating *C. albicans* cells with EDTA. Also, we appreciate your comment that the manuscript is sound and well-presented. However, we are afraid that the respected reviewer assumed the CAET cells as dead cells while they only divide relatively slower than the untreated cells. In the revised manuscript, we have presented additional evidence to show that CAET are live cells (Supp. Figs 2) and based on the new data, we expect a positive change in the reviewer’s opinion. Since CAET is a live strain, the data presented here is novel.

**Reviewer #2 (Public Review):**
Summary:Invasive fungal infections are very difficult to treat with limited drug options. With the increasing concern of drug resistance, developing an antifungal vaccine is a high priority. In this study, the authors studied the metal metabolism in *Candida albicans* by testing some chelators, including EDTA, to block the metal acquisition and metabolism by the fungus. Interestingly, they found EDTAtreated yeast cells grew poorly in vitro and non-pathogenic in vivo in a murine model. Mice immunized by EDTA-treated Candida (CAET) were protected against challenge with wild-type Candida cells. RNA-Seq analysis to survey the gene expression profile in response to EDTA treatment in vitro revealed upregulation of genes in metal homeostasis and downregulation of ribosome biogenesis. They also revealed an induction of both pro- and anti-inflammatory cytokines involved in Th1, Th2 and Th17 host immune response in response to CAET immunization. Overall, this is an interesting study with translational potential.Strengths:The main strength of the report is that the authors identified a potential whole-cell live vaccine strain that can provide full protection against candidiasis. Abundant data both on in vitro phenotype, gene expression profile, and host immune response have been presented.Weaknesses:A weakness is that the immune mechanism of CAET-mediated host protection remains unclear. The immune data is somewhat confusing. The authors only checked cytokines and chemokines in blood. The immune response in infected tissues and antibody response may be investigated.

Reply: Thank you very much for appreciating our work and finding our strain to be a live whole-cell anti-fungal vaccine strain with translational potential. Since the current study focused on the identification and detailed characterizations of a non-genetically modified live-attenuated strain and determination of its safety and efficacy as a potential vaccine candidate in the preclinical model, we have excluded the possible immune mechanisms involving CAET. In a separate study, we are currently investigating both cellular and molecular mechanisms that provide protective immunity in CAET-vaccinated mice.

**Reviewer #3 (Public Review):**
Summary:The authors are trying to find a vaccine solution for invasive candidiasis.Strengths:The testing of the antifungal activity of EDTA on Candida is not new as many other papers have examined this effect. The novelty here is the use of this EDTA-treated strain as a vaccine to protect against a secondary challenge with wild-type Candida.Weaknesses:However, data presented in Figure 5 and Figure 6 are not convincing and need further experimental controls and analysis as the authors do not show a time-dependent effect on the CFU of their vaccine formulation. The methodology used is also an issue. As it stands, the impact is minor.

Reply: Thank you so much for appreciating our efforts to develop a novel vaccine against fungal infections. We are extremely sorry for the lack of clarity in our writing related to Figs. 5 and 6, we have now modified the text and hope that the respected reviewer will find these convincing.

**Recommendations for the authors:**
Although the reviewers recognize the importance of the manuscript, they would like to see: (1) comparisons between their whole-cell vaccine and others tried in the field, (2) an investigation of the immune response in infected tissues and antibody response, and (3) more controls in Figures 5 and 6, and a time-dependent effect on the colony-forming units of their vaccine formulation. Please, address the questions and submit a revised version together with a rebuttal letter addressing point-by-point raised by each reviewer.

Reply: (1) We are afraid that a comparative study of a live and heat-killed cell vaccines will mislead the information presented here. This is the only non-genetically modified antifungal vaccine candidate therefore a comparison with a dead strain at present is unwarranted. We have now added supporting data to confirm that, the survivability of *C. albicans* cells was unaffected at 6 hr of EDTA treatment (CAET, Supp. Fig. S2). (2) Since the current study focused on the identification and a detailed characterization of a non-genetically modified live attenuated strain and its safety and efficacy as a potential vaccine candidate in the preclinical model, we have excluded the possible immune mechanisms involving CAET. However, in a separate study, we are currently investigating both cellular and molecular mechanisms that provide protective immunity in CAET-vaccinated mice. (3) The results of Figs 5 and 6 were misinterpreted by the respected reviewer, please see the explanation below.

**Reviewer #1 (Recommendations For The Authors):**
Some specific comments/suggestions for the authors:(1) What was the viability of the yeast after EDTA treatment? Is the delayed growth response because many cells died and it takes a while for remaining viable cells to catch up? This is important to know because it may mean the dose given to mice is substantially different and that should be accounted for. Some PI staining of the cells after treatment would help.

Reply: The growth curve assays (Fig. 1A and 1E) were initiated with O.D.600nm=0.5 of each cultures (~ 107 cells/mL) and the analyses suggested that the EDTA-treated *C. albicans* cells grew slower than the untreated cells. Fig. 1B and 1F further demonstrated that EDTA has minimal effect on the survival of the strain up to 8 hrs post-exposure. The proportion of the number of cells increased without and with metal chelators almost remained the same for this duration (0 – 8 hrs). Therefore, for subsequent analyses, 6 hr treatment was selected and such treated cells were considered as CAET, which were actively dividing live cells, albeit slower than untreated cells. As suggested and to strengthen our finding, a time dependent SYTOX Green and Propidium iodide staining of *C. albicans* cells without and with EDTA treatment was carried out and analysed by flow cytometry and microscopy, respectively. Both analyses revealed that the percentage of dead cells up to 12 hrs of without and with EDTA treatment remained the same. The new data has now been added in the revised version of the manuscript as Figure 1—figure supplement 2.

(2) In line with the above, what was the viability of the CAET cells after 3h in media? In the macrophage in vitro experiments, how do you know the reduced viability of the CAET cells is macrophage-specific? Did you run a control of CAET cells in media on their own to determine how CFU changed in macrophage-free conditions? Is the proliferation rates of untreated and CAET cells different? That would affect CFSE labelling and results. These experiments would work better with a GFP-expressing *C. albicans* strain, which is widely available. In the images in Figure 4c, it looks like there are more hyphae in CAET than untreated - was hyphal induction checked/measured? That's important to know because more hyphae usually means more clumping and this can affect CFU counts (giving the impression of less CFU when actually there is more). Because of all the issues above, I'm not fully convinced by the uptake/killing data.

Reply: As explained in response 1, we used actively dividing WT and CAET cells, and equal number of these cells were CFSE labelled. As can be seen in Fig.4A, the rate of phagocytosis was the same in 1 hr of pre-culture, but in the subsequent time points the double-positive cells were reduced in the case of CAET cells and that is due to fungal killing by macrophages. Fungal cells were released from the macrophages by warm water treatment and CFU was determined. Fig. 4B suggested that at 1hr of co-culture, the CFU of both fungal cells (WT and CAET) were the same and the fungal clearance was observed at later time points. Thus, the reduced viability of CAET cells was macrophagespecific. EDTA has minimal effect on hyphal transition without and with the presence of serum and the new data has now been provided in the revised version (Figure 1—figure supplement 3).

**Author response image 2. sa3fig2:** 

(3) Pooled data should be shown for all animal experiments.

Reply: Thank you for the suggestion, wherever it was meaningful pooled data for the animal experiments have now been provided.

(4) Immune cell counts/analysis in the kidney and bone marrow would be hugely helpful and more relevant to understanding immune responses following immunisation/infection. I think a more interesting analysis for the authors to consider would be to immunise with heat-killed yeast vs EDTAtreated yeast and see if there is a qualitative difference or better protection, i.e. is the EDTA-treated whole-cell vaccine superior to the heat-killed version? That is a better question to address. As it stands, the data in the paper is not surprising.

Reply: The studies on cellular and molecular mechanisms underlying protective immunity in CAETvaccinated mice are under progress in a separate study. This study mostly focused on the identification and detailed characterization of a non-genetically modified live-attenuated strain and its safety and efficacy as a potential vaccine candidate in a preclinical model. We are afraid that a comparison of a live cell (CAET) with a dead cell (heat-killed) will dilute the content of the manuscript and will not be meaningful. It is well accepted that the heat-killed *C. albicans* strain only provides partial short-lived protection to re-challenge (Refs-PMIDs: 12146759, and 9916097), thus, it does not warrant any comparison with CAET.

**Reviewer #2 (Recommendations For The Authors):**
Overall, this is a highly interesting study. I have the following specific comments for clarification.(1) In the introduction, the authors mentioned other anti-candida vaccines that are mostly effective against Candida infection by inducing neutralizing antibodies. However, in their CAET vaccine candidate, they only checked the cellular immunity in blood and found a balanced immune response (both pro- and anti-inflammatory responses are induced). How about the antibody production in these mice? It is a bit surprising that both untreated Candida infection and CAET Candida infection produced similar immune activation based on Figure 6, yet the CAET immunization provides protection. Some innate cell recruitment is higher in untreated Ca infection than the CAET infected mice (Figure 5F). The overall results on immune response characterization did not seem to explain why the CAET infection led to host protection while untreated Ca infection cannot. Characterizing infected tissue immune cell differentiation and cytokine production may offer some additional insights.

Reply: We agree with you that in this manuscript we have not provided any mechanistic study on the protective immunity in CAET-vaccinated mice. This will be demonstrated in a subsequent study.

(2) In Figure 5, some critical data seem to be missing in panels B and C. The CFU and histopathological images for CAET-treated mice challenged by Ca should also be shown there for comparison. Although they did show some data in Figure 5E and Figure S4, it is necessary to have that data in 5B and 5C from the same experiment. Figure S4 is a very busy figure and the images are quite small. It may be necessary to use arrows to point out what information authors want to emphasize.

Reply: Fig 5 B and 5C showed the data for mice that succumbed to infection. Since the other mice (saline control groups, CAET infected, CAET vaccinated, and re-challenged groups) survived, they were not sacrificed; therefore, the CFU data was not collected. In addition, we wanted to see the longevity of these survived mice and after 1 year of observations, they were handed over to the animal house for clearance as per the institutional guidelines. However, Figure 5E and Figure S4 (now Fig. S6) included all the mice groups as they were sacrificed at various time points irrespective of humane end points. As suggested FigS6 has now been modified and fungal cells were denoted by yellow arrows.

(3) EDTA-treated yeast cells showed poor growth but also had thicker cell walls with high chitin, glucan, and mannan levels. What leads to its clearance in vivo remains unclear, as usually, cells with thick cell wall structures and low metabolism are more resistant to stress, e.g., dormant cells. Macrophages were shown to contribute to CAET killing in a phagocytosis assay (Figure 4). Checking cytokines produced by macrophages during co-incubation may offer some insights. In all, additional discussion on what caused in vivo clearance would be helpful.

Reply: Mechanistic study on the protective immune responses of CAET will be demonstrated in a separate study. As suggested, the discussion section now contains additional information emphasising the in vivo clearance of CAET cells in the 3rd paragraph of discussion section.

(4) Long paragraphs in the discussion section could be divided into a bigger number of shorter paragraphs.

Reply: Thank you for the suggestion, it has now been modified in the revised version (7 short paragraphs). To make it more comprehensive, some of the content has been removed.

**Reviewer #3 (Recommendations For The Authors):**
(1) It is unclear how many cells were treated with 250 micromolar of EDTA for 6 hours before preparing the inoculum. It seems that only the OD was measured before adding EDTA. This is not a very rigorous and reproducible method.

Reply: In this manuscript, we have repeatedly used the same protocol to generate CAET cells for various analyses. The O.D.600nm = 0.5 culture is equivalent to 107 *C. albicans* cells per mL and this information has now been added in the revised manuscript.

(2) Upon treatment with 250 micromolar of EDTA, cells were harvested and counted to prepare the inoculum (5x10e5) for injecting it in mice. However, it appears that CFU of the inoculum was not done. Based on data shown in Fig. 1B, 250 micromolar of EDTA does inhibit Candida cell replication. Thus, the authors may have counted dead cells and, thus, injected dead cells together with live cells for the CAET inoculum. Thus, mice receiving this inoculum may have been infected (and vaccinated) with a lower number of live Candida cells.

Reply: Please see a similar response to reviewer #1. EDTA has minimal effect on the survival of *C. albicans* cells at 6 hr (also see supp. Fig. S2). We have already mentioned the CFU analysis of untreated and CAET cells in the methodology section related to inoculum preparation.

(3) It is unclear if 6 hours of treatment with 250 micromolar of EDTA is enough to induce a block of Candida cell replication. In Figure 1B, the authors treated for 24h. The authors are encouraged to wash the cells after 6 hours of treatment and see if their cell division will recover upon removal of EDTA.

Reply: Thank you for the suggestion. At 6 hr treatment, survivability of *C. albicans* cells was unaffected upon EDTA exposure. PI and SYTOX GREEN staining confirmed it (Supp. Fig. 2). Additionally, as suggested a rescue experiment was carried out by exogenous addition of divalent metals after 6 hr EDTA treatment and growth/CFU analyses were followed thereafter. A modified Fig. 1 A and B with new data has been provided.

(4) The data shown in Figure 5A is extremely exciting. However, the number of mice in each group (n=6) is too low. Normally, 10 mice per group are used for virulence studies unless the authors provide a power analysis that 6 mice per group will be sufficient. Also, CFU data were only provided for Ca and saline-Ca groups (Fig. 5B) and not for the other groups. CFU data should be provided for all mice.

Reply: Thank you for the suggestion and a statistical analysis of Fig. 5A was provided in the revised version. The rationale behind not including all mice groups in Fig. 5B is already explained in a response to reviewer #2.

(5) It is unclear how the authors differentiate between CFU arising from CAET or from WT Candida.

Reply: Since the Fig 5 E demonstrated that no CAET cells were detected in the kidney beyond 10 days of inoculation, in the re-challenged mice group (1CAET 2 Ca), the fungal cells those detected in the 3rd and 7th days were from the later inoculated cells (brown colour).

(6) Figure 5E: it is unclear if a 1 saline-2 saline (Figure legend) or if 1 saline-2 Ca (text) group was included. If the latter, where are the CFU? It is impossible that 1 saline-2 Ca mice have no CFU.

Reply: Thank you so much for pointing this out. The legend has now been modified that include 1saline-2saline and 1CAET-2Ca.

(7) It seems that CFU is significantly present in the kidney in the 1 CAET - 2 Ca group at day 7 but not at day 3. How is this possible? This is an extremely invasive model of infection, and the authors are challenging intravenously 500,000 live Candida cells. If by the 3rd day, the authors detect no CFU, then how is it possible that CFUs are arising on day 7?

Reply: We do detect fungal cells on 3rd day in 1CAET 2 WT mice group (~2000 cells), albeit much lower than in 7 days (~11200 cells). A Log10 scale graph has now been provided for better representation.

(8) Most importantly, if the authors are not detecting CFU at day 3, then earlier time points (e.g. day 2, day 1, or even 12 hours post-challenge) must be analyzed. The authors should show that CFU from the organs is decreasing in a time-dependent manner. Also, all CFU should be shown as Log10.

Reply: please see the previous response.

(9) Fig. 6: because it is unclear if the mice were challenged with the same inoculum of live Candida cells (untreated and treated with EDTA), the different cytokine profiles between the two groups could be simply due to the different inoculum sizes and not to the effect of EDTA on Ca.

Reply: please see the previous response as given also for Reviewer 1.